



# Sources, Reactivity and Burial of Organic Matter in East China Sea Sediments, as Indicated by a Multi-geochemical Proxy Approach

Xihua Yu[1,2], Shengkang Liang[1,2], Guicheng Zhang[1,2,3], Shanshan Li[1,2], Haifang Huang[1,2], Haoyang Ma[1,2]

[1]College of Chemistry and Chemical Engineering, Ocean University of China, Qingdao, 266100, China
[2]Frontiers Science Center for Deep Ocean Multispheres and Earth System, and Key Laboratory of Marine Chemistry Theory and Technology, Ministry of Education, Ocean University of China, Qingdao 266100, China
[3]Tianjin Key Laboratory of Marine Resources and Chemistry, Tianjin University of Science and Technology, Tianjin, 300457, China

*Correspondence to*: Shengkang Liang (liangsk@ouc.edu.cn)

**Abstract.** Large-river estuaries and adjacent seas play an important role in material exchanges and interactions in the land-ocean continuum and thereby impact global marine biogeochemistry. Due to the highly dynamic and complex transport and transformation processes of organic matter (OM), its distribution, sources, and reactivity in this region, especially under the multiple pressures of intense human activities and climate change, are not fully understood. An East China Sea transect,

ranging from the mouth of the Changjiang River to the Okinawa Trough (OT), was selected to investigate the sources and reactivity of OM in surface sediments using multi-geochemical proxies. Carbon (C) to nitrogen (N) ratios and stable isotopic signatures, total hydrolyzable amino acids, neutral sugars and organic carbon (OC)-normalized total lignin-phenol indicated that OM in nearshore surface sediments derived primarily from terrestrial vascular plants, while offshore OM was dominantly derived from marine production. In the estuary vegetation mainly consisted of gymnosperms, whereas nonwoody

angiosperms were dominant in offshore regions. Hydrodynamics, i.e., Changjiang Diluted Water and the Kuroshio Current markedly impacted sediment characteristics along this transect. The degree of OM degradation increased seaward, and sedimentation rates and OC burial fluxes were highest in Changjiang prodeltaic sediments and lowest in the OT. These results based on multi-biomarkers will advance our understanding of OC sources and burial during transport and deposition processes from estuaries to the deep oceans.

**Short summary**

Long-term preservation of organic carbon in marine sediments is a key process to buffer atmospheric carbon dioxide. Surface sediments from an East China Sea shelf transect were analyzed combining multiple approaches. Data shows that sedimentary organic carbon (SOC) distributions were controlled by primary production, currents and sediment type. Multi-proxies were consistent in indicating SOC sources. SOC degradation degree and burial rate increased and decreased seaward,

respectively.



## 1 Introduction

Global continental margins, accounting for more than 80% of global organic carbon (OC) burial (Guo et al., 2021b), are important sites of terrestrial organic carbon ($OC_{terr}$) deposition and marine organic carbon ($OC_{marine}$) in situ production. Over 90% of riverine carbon is buried in large-river delta-front estuaries (LDEs) and their adjacent ocean margins (Li et al., 2025; Yao et al., 2015; Zhao et al., 2024). Globally, ca. 170–200 Tg of particulate organic carbon (POC) is transported annually to the oceans by rivers, whereas only ca. 40–75 Tg OC of phosphorus (P) is preserved in sediments (Wang et al., 2022; Zhao et al., 2023). The fate of $OC_{terr}$ in the ocean has been referred to as a "geochemical conundrum"(Hedges et al., 1997). Therefore, understanding the carbon cycle, carbon organic matter (OM) sources, and their control by associated underlying processes, i.e., OM degradation and burial in estuaries and marginal seas, is critically important (Li et al., 2025; Yao et al., 2015).

Changes in sedimentary organic matter (SOM) have been estimated over the past few decades (Zhao et al., 2024). Before its final burial, SOM is exposed to long-term hydrological reworking in a variety of active zones, such as estuaries, beaches, and open shelves (Zhu et al., 2011). Organic carbon depositional processes and burial efficiency are governed by the physical characteristics of the sedimentary environment, mineral matrix, priming effect, etc. (Zhu et al., 2011). In particular, understanding of OM affected by hydrodynamic processes and the fate of OC upon burial is incomplete (Wang et al., 2022). The distribution of SOM in marginal seas is characterized by high spatio-temporal dynamics and heterogeneity due to the strong and complex land-sea interactions occurring in these marginal seas (Mei et al., 2019). These regions exhibit an increased incidence and intensity of droughts and floods due to climate change and anthropogenic disturbances, ultimately leading to considerable changes in sediment loads and riverine OM processing (Fernandes et al., 2020). The spatial extent of these impacts on OC circulation from continental margins to the deep sea remain unclear.

Bulk carbon to nitrogen (C/N) ratios can provide a rough indicator of the source of OM. A high C/N ratio (>20) and a low C/N ratio ranging from 4 to 10 are indicative of OM from terrestrial and marine origins, respectively (Ankit et al., 2022; Schmidt et al., 2009; Tesi et al., 2007). However, these distinctions can be blurred substantially by the increase of N-rich compounds during microbial (e.g., bacterial, fungal) lignin decay, the sorption of inorganic N on clay minerals, and the tendency of OM to lose more N than C during degradation by marine plankton (Tesi et al., 2007). Additionally, diagenesis, grain size, and many other factors may limit the accuracy of SOM C/N ratios in the identification of OM sources (Zhao et al., 2024). Stable isotope proxies, such as stable carbon ($\delta^{13}$C) and nitrogen isotope ($\delta^{15}$N) signatures, have been applied to trace OC sources (Duan et al., 2019). Thus, $\delta^{13}$C has been used to reliably differentiate between the OC origin from terrestrial plant material and marine phytoplankton (-24 to -18‰) (Duan et al., 2019; Martens et al., 2019). Isotopic $\delta^{13}$C further distinguishes among the terrestrial plant inputs of the Calvin-Benson cycle (C3) (-33 to -24‰), the Hatch-Slack cycle (C4) (-16 to -9‰), or the Crassulacean acid metabolism (CAM) cycle of vegetation (-20 to -10‰) (Liu et al., 2020; Wang et al., 2018a). Based on the $\delta^{13}$C values of suspended particulate matter (SPM) and sediments in the Ganga–Brahmaputra river system, C4 plants were identified as the main vascular plants in the delta region, relative to the plateau region (Li et al.,





2015). The two-end-member mixing model based solely on $\delta^{13}C$ is used to distinguish OC from terrestrial or marine origins

but could not further identify whether terrestrial OM originates from plants or soil. Currently, the three-end-member mixing model combining $\delta^{13}C$ and OC-normalized total lignin-phenol ($\Lambda8$) concentration is used to distinguish among terrestrial, marine, and sedimentary OC, but ignores the fact that the boundary between soil and terrestrial OM is not clearly defined. Moreover, $\delta^{13}C$ in marine sediments can be affected by many processes, such as the pre-deposition and degradation of OM during sediment transport, the input of recycled ancient carbon, and combined contributions from terrestrial C4 plant-derived

detritus (Sobrinho et al., 2021; Zhou et al., 2018a). Furthermore, the range of $\delta^{13}C$ variation is relatively small, typically only a few ppm. Therefore, it is difficult to precisely discern the origins of OM based on $\delta^{13}C$ alone because of the overlap in isotopic signatures. Values of $\delta^{15}N$ can be used to discriminate between marine (3–12‰) and terrestrial OM (around 0‰) (Gireeshkumar et al., 2013; Wang et al., 2018a). However, the $\delta^{15}N$ value of some marine phytoplankters such as *Trichodesmium* is also close to zero (Gireeshkumar et al., 2013). Additionally, kinetic isotopic fractionation affects $\delta^{13}C$ and

$\delta^{15}N$ values, and the sensitivity to diagenetic effects also limits their application.

Biomarkers such as lignin, total hydrolyzable amino acids (THAA), and neutral sugars (NS), can serve as powerful tools to trace OM biogenic sources and diagenetic transformations and have previously been used for the identification of OM sources (Liu et al., 2020; Zhang et al., 2019). Lignin-derived phenols (LPs), which are unique components of vascular terrestrial plants, are not readily degraded (Liu et al., 2020). Thus, indices involving these compounds, i.e., syringyl/vanillyl

(S/V) and cinnamyl/vanillyl (C/V) ratios and the lignin phenol vegetation index (LPVI) can provide information on paleo-vegetation and indicate changes in the composition of vegetation (Hu et al., 2012; Tareq et al., 2004), and the acid to aldehyde ratios of vanillyl (Ad/Al)v and syringyl (Ad/Al)s reflect the degree of lignin microbial degradation (Tareq et al., 2004). Unlike the total lignin concentration ($\Sigma8$), i.e., the sum of the concentrations of V, S, and C, the concentration of organic carbon (OC)-normalized total lignin-phenol ($\Lambda8$) directly reflects the content of terrestrial OM present (Tareq et al.,

2004). THAA and neutral sugars (NS), which are labile in nature, can reflect the degradation state of OM, although they play different roles in organisms (Fernandes et al., 2020). Generally, compared with bulk OM, THAA and NS are more readily degraded (Wei et al., 2021). Consequently, the reactivity of OM in marine sediments has been determined using the ratios of carbon-normalized neutral sugar yields (NS(%TOC)) and carbon-normalized amino acid yields (THAA(%TOC)) (Grutters et al., 2002; Kogel-Knabner, 2002; Wang et al., 2018b). Sediments typically undergo selective diagenesis, which is reflected in

lower values of THAA(%TOC) and NS(%TOC) (Fernandes et al., 2020). However, these molecular biomarkers represent only a portion of the bulk OM and can lead to potentially equivocal source assignation during biogeochemical processes. In particular, biomarker signals can be biased due to the dynamic nature of coastal environments (Zhu et al., 2008).

Multiple approaches have been combined in order to address the limitations of mono-proxy approaches. Whereas isotopic differences are sometimes not statistically significant between soil and woody materials, lignin can be used to resolve this

ambiguity (Tesi et al., 2007). A multiproxy approach was used to qualitatively compare different OC inputs (Yao et al., 2015). For example, the relative abundance of NS, the C/N ratio, and $\delta^{13}C$ strongly suggested that terrestrial OC underwent intensive degradation in the Pearl River estuary (He et al., 2010). Two or more biomarkers were combined and applied in a





few studies (Davis et al., 2009; Sobrinho et al., 2021), yet multiple biomarkers have rarely been used to reveal biogeochemical processes of OM in conjunction with hydrodynamics.

The East China Sea (ECS), one of the largest continental shelves worldwide, receives large nutrient inputs from the Changjiang River (CJR) and is affected by several currents. However, it appears that the transport of terrestrial material to the ECS outer shelf by the cross-shelf current has a limited influence on local sediment dynamics (Wang et al., 2008). In addition, sediment supply to the ECS has decreased greatly over the last two decades due to dam emplacement, which affects the delivery of $OC_{terr}$ (Wang et al., 2022; Wu et al., 2013). Previous studies have considered the role of nutrient cycling and

particle OM transport in the inner shelf region and in the water column, yet fewer studies have quantified the different kinds of OC burial flux and considered the role of the Kuroshio Current (KC) in determining these fluxes (Diesing et al., 2024; Wang et al., 2008; Xing et al., 2011; Zhu et al., 2011). The spatial variability and degree of degradation of both terrestrial and marine OM in sediments along a large cross-section from the Changjiang Estuary (CJE) to the Okinawa Trough (OT) are still poorly understood (Wang et al., 2008; Xing et al., 2011). Additionally, few studies have linked OC sediment data

and hydrodynamics using a multi-geochemical proxy approach to determine the coastal distribution of OM.

    In the present study, the source, transport, and reactivity of OM in surface sediments of a transect from the CJE to the OT were determined using a variety of proxies. The selected transect provides an ideal area to investigate the hydrodynamic-driven redistribution of OC. Thus, key objectives of this study were to compare the sources, degradation, and burial of OM using multiple proxies, thereby providing a basis for enhanced understanding of the OC cycle and the role of hydrological

dynamics in large-river estuaries and continental shelves in global carbon cycling.

## 2 Materials and Methods

### 2.1 Study area and sample collection

The Jiangsu Coastal Current (JSCC), Zhejiang-Fujian Coastal Current (ZFCC), Yellow Sea Coastal Current (YSCC), Changjiang Diluted Water (CDW), Taiwan Warm Current Water (TWC), KC, and Shelf Mixed Water (SMW), which is

created by combining several water masses, are all included in the ECS (Fig. 1) (Guo et al., 2018). River runoff and human activity provide allochthonous inputs to the ECS, while phytoplankton primary productivity provides autochthonous inputs. DOC and POC from the ECS shelf are exported to the western Pacific by the KC, the main current connecting the Pacific and the ECS (Bauer et al., 1998; Liu et al., 2003). The cross-shelf study transect is located from 30.5° to 31.5° N and from 122.3° to 128.2° E, with water depths ranging from 17 to 519 m. This transect is characterized by complex hydrological

dynamics and includes the TWC, KC, and CDW. A total of eight surface (0–2 cm) sediment samples were collected along the transect from the delta to the OT on board the R/V *Dong Fang Hong 2* and *Science 3* scientific research vessels on May 2009 and June 2010 (Fig.1, Table 1, and Table S1). Samples at stations C0501, CFJA, C0508, and C0608 were collected on May 2009, and those at stations DHa-2, DHa-3, DHa-4, and DHa-5 were collected on June 2010. The surface, middle and bottom water column physical, chemical, and biological parameters measured at the sampling sites are shown in Tables S1




and S2. Sediment samples were divided into three representative groups based on water depth and sediment type (Bao et al., 2019): (Ⅰ) prodeltaic sediments at stations C0501 and CFJA which were sampled at depths <30 m, mainly consisted of silt and clay at the surface; (Ⅱ) shelf sediments at DHa-2, DHa-3, DHa-4, DHa-5, and C0508, which were sampled at depths ranging from 30 m to 160 m, and (Ⅲ) OT station C0608, collected at depths generally >600 m, which consisted of silt to clay. Samples were kept at -20° C on board the vessels.


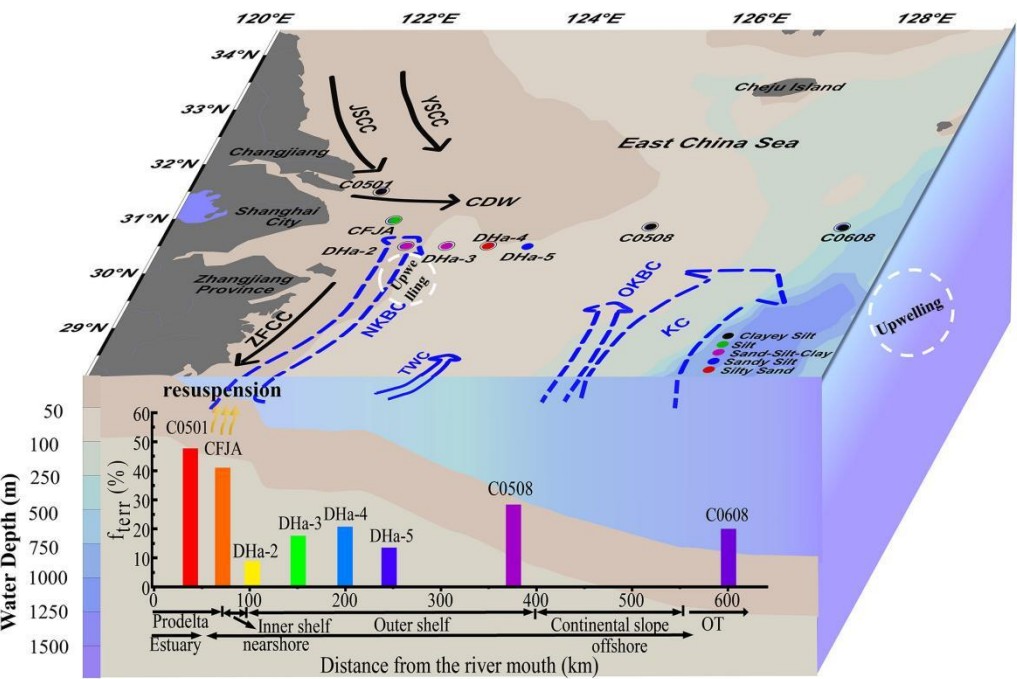

**Figure 1. Diagram of sampling stations in the study transect and regional ocean circulation patterns (black solid lines and dashed blue lines). KC, Kuroshio Current; TWC, Taiwan Warm Current; JSCC, Jiangsu Coastal Current; YSCC, Yellow Sea Coastal Current; ZFCC, Zhejiang-Fujian Coastal Current; CDW, Changjiang Diluted Water; OKBC, offshore Kuroshio Branch Current;**
**NKBC, nearshore Kuroshio Branch Current. Landforms at the continental margins are classified according to water depth. fterr = percentage of carbon from terrestrial sources.**

## 2.2 Sample pretreatment and analysis

Sediment samples were freeze-dried, and OM and calcium carbonate were first removed by hydrogen peroxide ($H_2O_2$) and hydrochloric acid (HCl), respectively. Next, they were dispersed and analyzed via a laser particle size analyzer (Beckman

Coulter Ltd., UK) to determine grain size composition. Inorganic C was removed before total organic carbon (TOC) and total nitrogen (TN) content determination using a Flash 2000 elemental analyzer (Thermo Fisher Scientific Inc., US) (Tesi et al., 2007). Isotopic values $\delta^{13}C$ and $\delta^{15}N$ were measured via the isotope ratio mass spectrometer MAT 253 (Thermo Fisher Scientific Inc., US) (Tesi et al., 2007). Values of $\delta^{13}C$ (‰) and $\delta^{15}N$ (‰) were calculated as follows (Wang et al., 2021):



$$\delta^{13}C, \delta^{15}N(‰) = \left(R_{sample}/R_{standard} - -1\right) \times 1000‰ \tag{1}$$

where the isotopic $^{13}C/^{12}C$ ($^{15}N/^{14}N$) ratios of the sample and standard are expressed by $R_{sample}$ and $R_{standard}$, respectively. The relative standard deviations (SD) of $\delta^{13}C$ and $\delta^{15}N$ were <1.0 ‰ and <0.05 %, respectively.

Lignin analysis was carried out using the alkaline cupric oxide (CuO) oxidation method (Hu et al., 2012; Jeunon Gontijo et al., 2021; Tareq et al., 2004). A Teflon reaction vessel was loaded with the samples, powdered CuO, and ammonium iron sulfate hexahydrate ($Fe(NH_4)_2(SO_4)_2 \cdot 6H_2O$) in aqueous NaOH solution. The vessel was placed at 170° C for 3 h and was

then immediately cooled. Then, 11 phenols formed from lignin oxidation were analyzed via a Shimadzu 2010 Plus gas chromatography system (Shimadzu Co., LTD., Japan) with on-column derivatization (Hu et al., 2012; Jeunon Gontijo et al., 2021; Tareq et al., 2004). Trans-cinnamic acid and ethyl vanillin were added to every sample to monitor the recoveries of lignin phenols. The recovery rates of lignin phenol surrogates were 64.72-103.23% for trans-cinnamic acid and 53.81-86.73% for ethyl vanillin. All lignin phenol data were corrected by the surrogate recoveries.

Analysis of THAAs was carried out by an Agilent 1260 high-performance liquid chromatography (HPLC) (Agilent Technologies, Inc., Germany) (Wei et al., 2021). Sediments were hydrolyzed using 6 M HCl at 110° C. Then, 14 amino acids were separated and quantified (Guo et al., 2021a). The amino acid degradation index ($DI_{AA}$) was calculated via principal component analysis (PCA) based on the molar percentage of various amino acids (Dauwe et al., 1999). The relative SD was <6.00%.

Analysis of NS used a Dionex 500 ion chromatography system (Thermo Fisher Scientific Inc., USA) (Skoog and Benner, 1997). Sulphuric acid (12 mol·L$^{-1}$) was added to freeze-dried and homogenized samples, followed by hydrolysis at 100° C for 3 h. After that, Milli-Q water was used to dilute the samples. The deoxy sugar (200 nmol·L$^{-1}$) was added as an internal standard. The sample was then run immediately through a mixed bed of exchange resins and injected into the chromatography system (Skoog and Benner, 1997). Average NS recovery was 72-107%, and the relative SD was <3.7% (n =

170 13).

A Seabird 911 conductivity-temperature-depth (CTD) apparatus (Sea-bird Scientific, USA) was used to determine local water temperature, salinity, and depth (Yang et al., 2011). Water column turbidity and chlorophyll *a* (Chl-*a*) concentration were measured with a YSI sonde. Water samples were collected by Niskin bottles and then filtered through Whatman GF/F glass fiber filters. The filter and filtrate were used for Chl-*a* and nutrients analysis, respectively (Yang et al., 2011).

**2.3 Indicators of organic matter sources**

A two-end-member mixing model was used twice to determine OM sources along the transect. The primary two-end-member mixing model discriminated between terrestrial (terr) and marine origins based on $\delta^{13}C$ values (Zhou et al., 2018a). N/C ratios were not considered as the end-member, due to their nonconservative behavior (Yao et al., 2015). Multiple sources of terrestrial OM were not considered by this mixing model. The second two-end-member mixing model further

distinguished between terrestrial OM sources (plant and non-plant OM) via Λ8 tracers (Zhang et al., 2014). Thus:





$$X_{sample} = X_{marine} \times f_{marine} + X_{terr} \times f_{terr} \tag{2}$$

$$f_{marine} + f_{terr} = 1 \tag{3}$$

$$f_{terr} = (X_{marine} - X_{sample})/(X_{marine} - X_{terr}) \tag{4}$$

$$f_{plant} + f_{terr-plant} = f_{terr} \tag{5}$$

$$f_{plant} = (X_{terr-plant} - X_{sample})/(X_{terr-plant} - X_{plant}) \tag{6}$$

TOC can be separated as $OC_{marine}$, $OC_{terr-plant}$, and $OC_{plant}$; their relationships are as follows:

$$OC_x = \text{TOC} \times f_x \tag{7}$$

$$OC_{plant} + OC_{terr-plant} = OC_{terr} \tag{8}$$

$$OC_{terr} + OC_{marine} = \text{TOC} \tag{9}$$

where $X_{marine}$, $X_{terr}$ and $X_{sample}$ are the δ¹³C values of the marine and terrestrial end-members, and the sediment sample, respectively. The terms $f_{terr}$ and $f_{marine}$ represent the percentages of C from marine and terrestrial sources, respectively, x represents soil, plant, and marine sources of OC, and $f_{plant}$ and $f_{terr-plant}$ represent the percentages of plant and non-plant terrestrial OM, respectively. The end-member values were derived from actual measurements from sources along the Changjiang region to ensure that they were appropriate for the study. The greatest POC enrichment according to δ¹³C values in ECS surface water was -20.1‰ and thus close to the global mean of -20.0‰ (Wang et al., 2021; Zhu et al., 2013). Therefore, -20.0‰ was selected as the marine end-member (Wang et al., 2021; Zhu et al., 2013). Prior results revealed that the average δ¹³C value of the CJR (range = -26.6 to -24.4‰) was -25.6‰ (Wang et al., 2021; Zhu et al., 2013). According to recent findings (Li et al., 2025), the average δ¹³C end-member value in the study area was −27.2‰ so it was selected as the terrestrial end-member for the present study's calculations. The Λ8 value of the plant end-member was 6.0 mg/(100 mg OC) according to results for plants in the Changjiang drainage basin (Bao et al., 2014). Since lignin is only found in vascular plants (and red algae, some Coleochaetale green algal species and some bryophytes) (Novo-Uzal et al., 2012), the Λ8 value of the non-plant end-member was 0.00 mg/(100 mg OC).

### 2.4 Average carbon burial rate

Sedimentation rate (SR) data were obtained from an earlier study (Table S3) (Guo et al., 2021b). Based on these authors' large and accurate dataset, the kriging interpolation method was applied to the target station (Guo et al., 2021b). The organic carbon burial flux ($F_{burial}$, g C·m⁻²·yr⁻¹) was calculated as follows (Song et al., 2016):

$$F_{burial} = C \times SR \times \rho \times (1 - \Phi) \tag{10}$$



$$F_x = F_{burial} \times f_x \tag{11}$$

where $C$ represents the measured TOC content in surface sediments (g C·g⁻¹), ρ is the sediment dry density (g·cm⁻³), $\Phi$ is

sediment porosity, and $SR$ represents the linear sedimentation rate (cm·yr⁻¹). Based on the literature, values of $\Phi$ and ρ are

0.75 and 1.2 g·cm⁻³, respectively (Guo et al., 2021b; Liu et al., 2007; Wang et al., 2017).

**2.5 Statistical analysis**

Statistical analysis was conducted using IBM SPSS Statistics 26 software. SR was treated by Surfer 15 with the kriging

interpolation method. The number of grid nodes in both the X and Y directions was set to 1000. Principal component

analysis was employed to identify differences between sample types. Origin 2021 was used for PCA, including 13 variables:

N/C, THAA(%TOC), NS(%TOC), Λ8, and so on. In addition, Origin 2021 software was used for all graphics.

**3. Results**

**3.1 Bulk parameters**

**3.1.1 Sediment grain size**

Grain size and sand content increased from the prodeltaic sediments (station C0501) to the outer shelf sediments (DHa-4),

and then decreased from station DHa-4 to the OT sediments (C0608); the mean ± SD grain size (range = 9.9–32.8 μm) was

16.5 ± 0.5 μm (Table 1). Surface sediment types along the transect from the estuary area to the OT ranged from silty to

sandy and then to muddy (Fig. 1). Silt and silty clay were distributed mainly in the CJE and the outer shelf current mud belt.

Grain size exhibited a positive correlation with the sand content in surface sediments (Table 1 and Fig. S1). At station DHa-4,

the sand content and median grain size reached their maximum value among all sampled sites (Table 1).




**Table 1. Sampling stations, water column depth, grain size characteristics, bulk sediment parameters, and biomarkers used (see text for definition) in the East China Sea transect undertaken in this study. TOC: total organic carbon; TN: total nitrogen; SR: sedimentation rate; Σ8: total lignin concentration; THAA(%TN): nitrogen-normalized concentration of total hydrolysable amino acids (see sec. 2 for calculation of F parameters; terr: terrestrial).**

| Station | Water depth (m) | Sand (>64 μm, %) | Silt (4 μm~64 μm, %) | Clay (<4 μm, %) | Median grain size (μm) | Average grain size (μm) | TOC (%) | TN (%) | SR (cm·yr⁻¹) | $F_{burial}$ (g C·m⁻²·yr⁻¹) | $F_{plant}$ (g C·m⁻²·yr⁻¹) | $F_{terr-plant}$ (g C·m⁻²·yr⁻¹) | $F_{marine}$ (g C·m⁻²·yr⁻¹) | Σ8 (mg/10gdw) | THAA (%TN) |
|---|---|---|---|---|---|---|---|---|---|---|---|---|---|---|---|
| C0501 | 17.0 | 1.7 | 74.7 | 23.7 | 11.6 | 9.9 | 0.70 | 0.068 | 1. | 28.9 | 5.3 | 8.5 | 15.1 | 1.6 | 20.6 |
| CFJA | - | 1.1 | 76.8 | 22.1 | 12.2 | 9.9 | 0.44 | 0.065 | 2.41 | 31.4 | 2.1 | 10.8 | 18.5 | 0.4 | 18.8 |
| DHa-2 | 58.1 | 20.8 | 54.03 | 25.2 | 11.5 | 13.2 | 0.40 | 0.064 | 2.14 | 25.7 | 0.2 | 2.1 | 23.4 | 0.2 | 17.1 |
| DHa-3 | 57.8 | 33.4 | 44.8 | 21.8 | 17.0 | 18.2 | 0.45 | 0.082 | 1.51 | 20.4 | 0.2 | 3.4 | 16.8 | 0.2 | 10.3 |
| DHa-4 | 56.0 | 46.5 | 38.5 | 15.0 | 53.7 | 32.8 | 0.39 | 0.079 | 1.23 | 14.3 | 0.06 | 2.9 | 11.3 | 0.05 | 11.5 |
| DHa-5 | 59.9 | 38.7 | 42.9 | 18.4 | 29.7 | 24.0 | 0.23 | 0.026 | 0.94 | 6.6 | 0.02 | 0.87 | 5.7 | 0.04 | 30.2 |
| C0508 | 75.0 | 16.9 | 56.2 | 26.9 | 8.8 | 11.4 | 0.33 | 0.062 | 0.46 | 4.4 | 0.04 | 1.2 | 3.2 | 0.06 | 25.9 |
| C0608 | 519.0 | 17.9 | 55.6 | 26.5 | 13.0 | 12.8 | 1.73 | 0.210 | 0.04 | 2.2 | 0.00 | 0.44 | 1.8 | 0.05 | 14.4 |

Note. -: Unknown.

### 3.1.2 TOC and TN content and ratios

The TOC content ranged from 0.23 to 1.73% and decreased from nearshore to offshore regions. The TN content exhibited a similar variation (Table 1), but the values of TOC and TN of the OT (station C0608) attained the maximum transect values. TOC adhered strongly to fine particles, and the TOC trend exhibited a lag relative to the clay content in surface sediments (Fig. S2). The atomic C/N ratio ranged from 5.7 to 12.0 and decreased offshore except at the outer shelf and OT stations (DHa-5 and C0608, respectively; Fig. 2(a)), whereas the C/N ratios at these two stations were higher than those of adjacent stations. Sedimentation rate ranged from 0.04 to 2.4 cm·yr⁻¹, and $F_{burial}$ ranged from 2.2 to 31.4 g C·m⁻²·yr⁻¹. Both parameters showed decreasing trends with increasing seaward distance from the CJE. Stations C0501 and CFJA in the prodelta had a relatively high SR and the highest silt and clay content. Calculated $F_{marine}$, $F_{terr-plant}$ and $F_{plant}$ ranged from



1.8 to 23.4, 0.4 to 10.8, and 0.0 to 5.3 g·C·m$^{-2}$·yr$^{-1}$, respectively (Table 1). The $OC_{terr-plant}$ and $OC_{plant}$ burial decreased along the study transect, while $OC_{marine}$ accumulated markedly in the OT (station C0608) (Table S1).

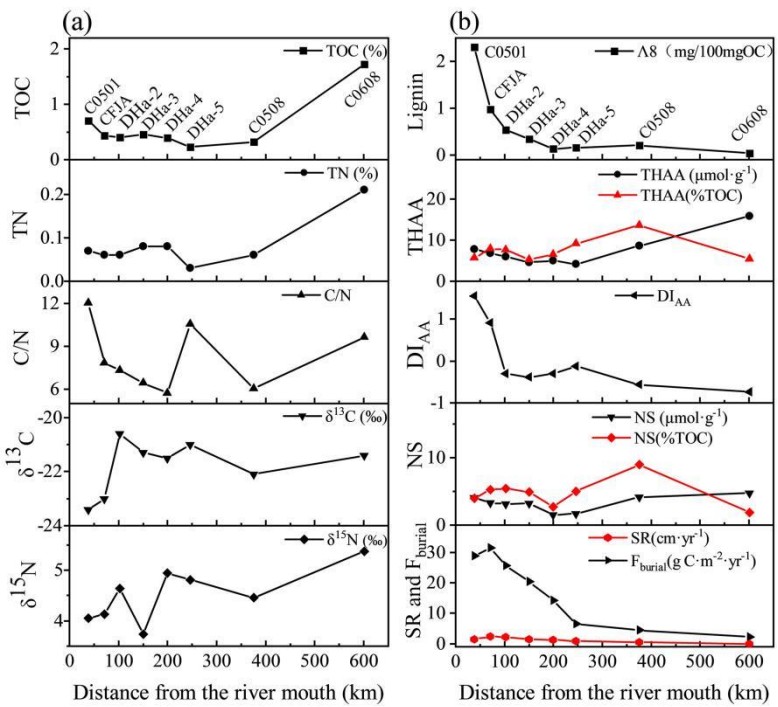

**Figure 2. (a) Spatial distribution of total organic carbon (TOC), total nitrogen (TN), C/N ratio, and isotopic signatures δ13C, and δ15N in surface sediments at the various indicated sampling stations; (b) OC-normalized total lignin-phenol (Λ8), total and carbon normalized concentration of hydrolysable amino acids (THAA and THAA (%TOC), respectively), and amino acid degradation index (DI$_{AA}$; see sec.2 for their calculation), neutral sugars (NS and NS (%TOC)) in the East China Sea transect.**

### 3.2 Carbon (δ$^{13}$C) and nitrogen (δ$^{15}$N) isotopic values

Carbon isotopic (δ$^{13}$C) values increased from -23.43‰ to -20.63‰, whereas δ$^{15}$N values ranged from 3.74‰ to 5.37‰ (except at station DHa-3) in surface sediments from the estuary towards the OT (Fig. 2(a)). According to the two-end-member mixing model, $f_{terr}$ values ranged from 8.8 (at station DHa-2) to 47.7% (at C0501) (Table S1).



### 3.3 Organic matter molecular composition and abundance

### 3.3.1 Lignin components and distribution

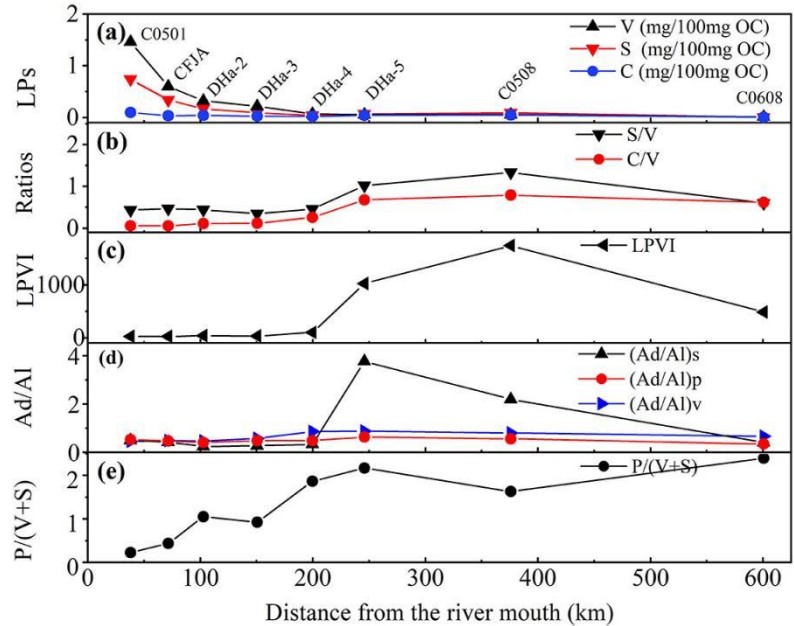

**Figure 3. Spatial distribution of lignin parameters in surface sediments at eight sampling stations: (a) lignin phenols, (b) lignin phenol ratios, (c) lignin phenol vegetation index (LPVI), (d) acid to aldehyde ratio of syringyl, p-hydroxyl and vanillyl (Ad/Al), (e) ratio of p-hydroxyl phenol to (vanillyl phenol + syringyl phenol) (P/(V+S)).**

The content of plant-derived phenols ($\Lambda 8$, $\Sigma 8$, S, V, and C) decreased markedly from nearshore to offshore regions (Figs. 2(b) and 3(a)), whereas those of the three lignin monomers decreased in the order: V (0.01–1.46 mg/100 mg OC) > S (0.01–0.74 mg/100 mg OC) > C (0.01–0.10 mg/100 mg OC) in surface sediments (Fig. 3(a)). Among them, V and S accounted for 70.0–96.6% of $\Lambda 8$, followed by C, which was consistent with the degradation rate of lignin phenols during soil humification (C > S > V) (Tareq et al., 2011; Tesi et al., 2007). Values of $\Lambda 8$ ranged from 0.03 to 2.31 mg/(100 mg OC), and the highest and lowest values appeared at the prodelta (C0501) and the outer shelf (C0608), respectively. $\Sigma 8$ ranged from 0.04 to 1.61 mg/g. Values of S/V, C/V, LPVI, (Ad/Al)v, (Ad/Al)s and P/(V+S) ranged from 0.35 to 1.34 (Fig. 3(b)), 0.06 to 0.79 (Fig. 3(b)), 24.0 to 1745.3 (Fig. 3(c)), 0.45 to 0.88 (Fig. 3(d)), 0.24 to 3.77 (Fig. 3(d)), and 0.23 to 2.40 (Fig. 3(e)), respectively. The LPVI, and S/V, and C/V ratios increased from the inshore station DHa-4 to the offshore station C0508. Furthermore, the value of (p-hydroxyl phenol/(vanillyl phenol + syringyl phenol) (P/(V+S)) increased from the estuary to the OT.



### 3.3.2 Neutral sugar composition and yield

The NS content decreased from the prodelta station (C0501) to the shelf station (DHa-4) but increased from DHa-4 to the OT station C0608 along the transect (Fig. 4(a)). Although NS and THAA yields generally followed the trends in TOC concentrations, the NS(%TOC) initially tended to increase from the prodelta station (C0501) to the nearshore station DHa-2 in the outer shelf, then decreased from DHa-2 to the offshore station (DHa-4) in the outer shelf, and finally increased from DHa-4 to the farthest offshore station (C0508) in the outer shelf (Fig. 2(b)). NS(%TOC) values varied greatly between 1.9 and 8.9%. The most abundant NS was glucose (Glc), followed by galactose (Gal), mannose (Man), D-arabinose (Ara), and D-xylose (Xyl) (Fig. 4(a)), while D-fucose (Fuc) and D-rhamnose (Rha) were minor components. The mol% Glc concentration decreased seaward, except at DHa-5 (Fig. 4(b)).

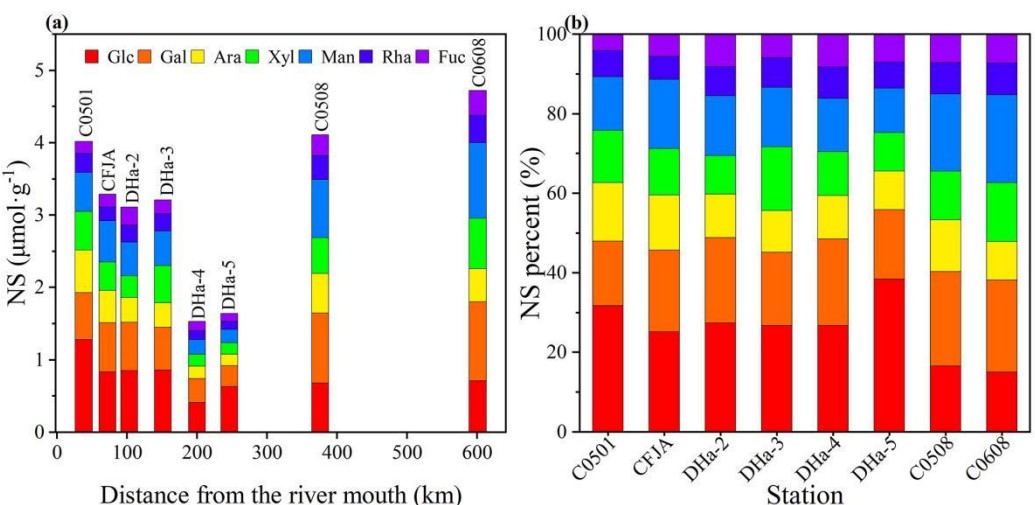

**Figure 4. Spatial distribution of neutral sugars (NS) in surface sediments (a) and their molar percentages (b) in surface sediments of eight sampling stations along the East China Sea transect. Glc: glucose; Gal: galactose; Ara: D-arabinose; Xyl: D-xylose; Man: mannose; Rha: D-rhamnose; Fuc: D-fucose.**

### 3.3.3 Amino acid composition and yield

The main components of THAA were glycine, aspartate, alanine, glutamic acid, and serine, which exhibited a similar pattern of first decreasing and then increasing values (Fig. 5(a)). Glycine exhibited the highest percent molar fraction (14.2–18.7 mol%) of the total molar concentration, and tyrosine had the lowest molar fraction (Fig. 5(b)). There was an obvious decreasing trend in THAA content from CJE to the outer shelf region (station DHa-5), and then the THAA concentration increased and was twice as high at the farthest outer shelf station C0508 than at DHa-5, and 3.8 x higher at OT station C0608 than at DHa-5 (Fig. 5(a)). The $DI_{AA}$ values decreased from 1.54 to -0.74 with increasing distance offshore (Fig. 2(b)). The values of THAA(%TOC) ranged from 4.7 to 13.5, increased from stations C0501 to CFJA in the prodeltaic region, decreased from station CFJA to DHa-3 in the outer shelf, and then increased from station DHa-3 to the farthest outer shelf (station




C0508), thus increasing with seaward distance (Fig. 2(b) and Table 1). Values of THAA(%TN) ranged from 10.3 to 30.2 and decreased from the C0501 station to the outer shelf (DHa-4) (Table 1). Although the maximum amino acid content was determined at the C0608 station, the percentages of THAA(%TOC) (5.4%) and THAA(%TN) (14.4%) were relatively low.

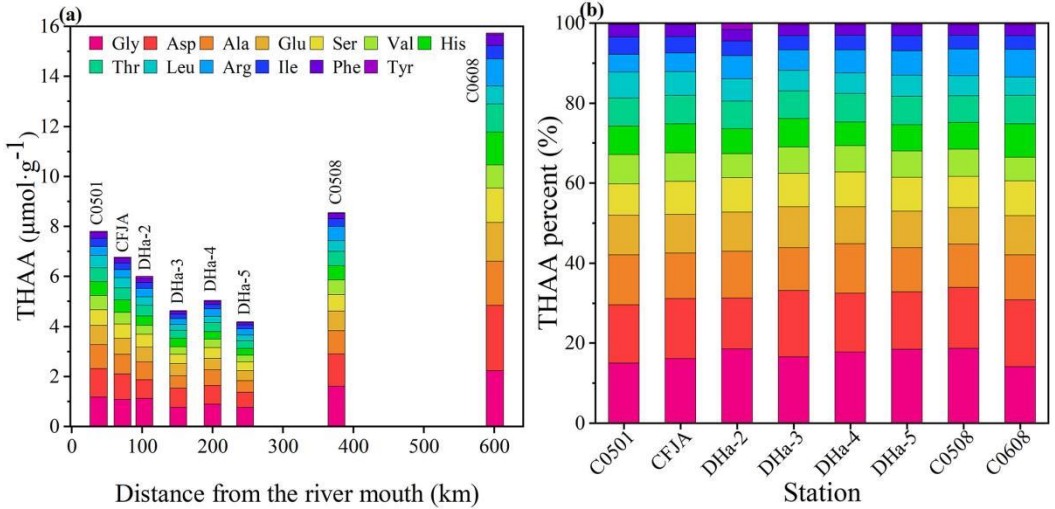

**Figure 5. Concentration of each individual hydrolysable amino acid (HAA) (a) and the molar percentages of individual HAAs out of the total (THAA) (b) in surface sediments at sampling stations in an East China Sea transect (see Table S2). Gly: glycine, Asp: aspartate, Ala: alanine, Ser: serine, Val: valanine, His: histidine, Thr: threonine, Leu: leucine, Arg: arginine, Ile: isoleucine, Phe: phenylalanine, and Tyr: tyrosine.**

### 3.4 Identification of OM sources and PCA estimation of relative OM source contributions

Principal component analysis performed using elemental NS, THAA, and lignin parameters in sediments and environmental conditions produced clear data separation. The variance of the first PC (PC1) and second (PC2) principal components accounted for 38.8% and 23.9%, respectively (Fig. 6). The former had high positive loadings for environmental conditions (distance from shore, depth, median grain size) and OC sources (LPVI, $\delta^{13}$C and $\delta^{15}$N). In contrast, PC2 exhibited strong positive loadings of sediment quality (NS(%TOC), THAA(%TOC)) and clay (Fig. 6). Stations DHa-4, DHa-5, C0508, and C0608 showed a positive relationship with PC1, while C0508 was positively related to PC2. The OM composition at offshore stations (DHa-4, DHa-5, C0508, and C0608) was markedly influenced by environmental conditions and OC sources, and the farthest offshore station (C0508) was markedly influenced by sediment quality (NS(%TOC), THAA(%TOC)), and clay content.





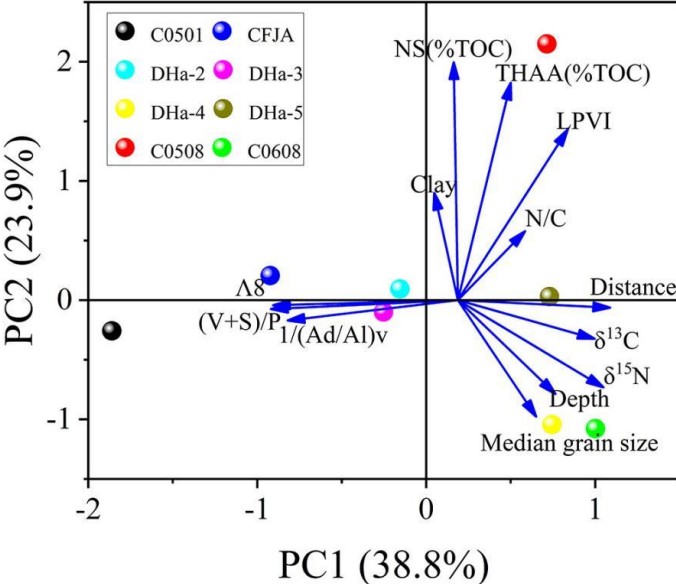

**Figure 6. Plot of variable loadings and sample scores obtained by principal component analysis (PCA) of Factors 1 and 2 of samples from surface sediments at eight sampling stations in the East China Sea. All variables and samples are indicated on the diagrams (see sec. 2 for abbreviations).**

## 4. Discussion

### 4.1 Sedimentary organic matter sources and distribution as indicated by multi-geochemical proxies

Multiple geochemical parameters were used in the present study to indicate the sources and distribution of OM from the estuary area to the OT. Higher TOC concentrations were determined in the prodeltaic region and OT sediments, where fine-grained sediments accumulated. This confirms that OC preferentially attaches to fine-grained particles (Mei et al., 2019). In the present study, the higher positive loading of THAA(%TOC), NS(%TOC), and clay on PC2 indicated that clay may be the main sink for THAA and NS. Furthermore, the TOC content was positively related to the silt and clay content along the

study transect (Fig. S2). A relatively low TOC content (0.39%) was found at station DHa-4, where coarse sediment contained the lowest clay and silt content (53.5%), which may be attributed to the limited accumulation of fine-grained particles. These results provide evidence of the important effect of sediment type on the distribution and content of SOM. The $\delta^{13}C$ values indicated that SOM in the prodelta (stations C0501 and CFJA), shelf (DHa-2, DHa-3, DHa-4, DHa-5, and C0508), and the OT (station C0608) was derived from marine sources. This result was consistent with that derived from the

C/N ratio. Additionally, C/N ratios indicated that the SOM of the prodeltaic station (C0501) was derived from mixed marine and terrestrial sources. $\delta^{15}N$ values also revealed that the SOM at these stations evolved from marine sources, based on a range from 3 to 12‰ for marine sources determined by Martens et al. (2019). Taken together, both the C/N ratio, and $\delta^{13}C$ and $\delta^{15}N$ values were thus consistent, indicating that SOM was derived mainly from marine sources with a decreasing




seaward trend in terrestrial OM except at stations C0508 and C0608, resulting from the replacement of marine OM and
progressive degradation of terrestrial OM (Fig. 2).

The present study demonstrates that the combined use of multi-indicators clearly improves the reliability of a single indicator. Generally, the three-end-member mixing model can discriminate among soil, plant, and marine OM sources, but in fact, soil and vegetation originate from terrestrial sources and their chosen end-members often appear isotopically indistinguishable, i.e., with mean values of -28.1 ± 1.7‰ and -26.3 ± 3.0 ‰ for C3 vascular plant and soil end-members,
respectively (Yao et al., 2015). The whole CJR drainage basin is dominated by OM from C3 plants (Zhu et al., 2011), thus resulting in their high estimated contribution to the degree of uncertainty (Arendt et al., 2015). A greater number of end-members in the end-member mixing model may introduce additional errors, so it is more reasonable to first divide OM sources into terrestrial and marine sources and then divide terrestrial OM sources into non-plant material (excluding vascular plants from terrestrial OM) and vascular plants. The calculation is more accurate using the two-end-member model twice
than by using the three-end-member model. Results of the two-end-member mixing model indicated that the $f_{terr}$ contribution to SOM in the western portion of the study area was greater than that in the eastern portion because of terrestrial OM from the CJR. The $f_{terr}$ term indicated that OM of prodeltaic sediments up to and including the OT sediments was mainly derived from marine sources. The $f_{terr}$ values in the Changjiang prodelta were intermediate between those in the Amazon River Delta (~30%) and those in the Bengal Fan (~100%) (Zhu et al., 2011; Zhu et al., 2008). The $f_{terr}$ values
(close to 50%) of prodeltaic sediments (stations C0501 and CFJA) indicated a primarily terrestrial input, which was consistent with PCA results of Λ8 near prodeltaic sediments (C0501 and CFJA) in Fig. 6. Organic matter of the CJR is transported via shelf advection and experiences rapid burial in floodplains and deltaic regions (Ramaswamy et al., 2008). Overall, the present study shows a decreasing trend of $OC_{terr}$ in vascular plant OM going seaward.

The unusual phenomenon at some stations requires further verification using combined water column parameters and
biomarkers. At DHa-2 in the outer shelf region (water depth = 50-160 m), $f_{marine}$ showed a pronounced increase (by 32.3% relative to that of CFJA), which could be attributed to rapid deposition during high-frequency algal bloom events (Table S1). Based on sampling time, and geographic location, the salinity and temperature at CFJA, DHa-2, and DHa-3 were compared to those at the corresponding long-term monitoring stations Ra1, Ra5 and Ra7a (Table S1 and S2). It was clear that the phosphate concentration in surface and bottom layers, the Chl-$a$ concentration in the middle layer, and salinity and
temperature in the bottom layer at DHa-2 (Tables S1 and S2) were higher than those in the corresponding water layers in the CFJA. In addition, salinity and temperature in the bottom layer of station DHa-2 were in the range of those in Kuroshio Subsurface water (Chen et al., 2021). Furthermore, the N:P ratio of 15:1 in DHa-2 bottom water mostly approximated the N:P Redfield ratio of 16:1. The Asp/Gly ratio and Ser+Thr concentrations were low and high, respectively, at DHa-2, (Fig. S3). This finding indicated that diatom blooms were caused by the influence of the nearshore Kuroshio Branch Current
(NKBC), which could also be confirmed by the enrichment of $^{13}$C in sediments and the Si/N (SiO$_3$-Si/dissolved inorganic nitrogen (DIN)) ratio of 2 in the water column (Table S2). Generally, diatom proliferation results from a Si/N ratio >1 (Wang,



2006). The NKBC is characterized by phosphate-rich bottom water from eastern Taiwan that flows northwestward along the 60 m isobath and finally reaches 30.5° N, turning east thereafter (Yang et al., 2012). These results indicated that the NKBC flowed through station DHa-2 and caused upwelling of deep water.

The offshore Kuroshio Branch Current (OKBC) promotes primary productivity (Wang et al., 2018c) and may result in input of some terrigenous OM into the shelf (C0508) and the OT (C0608) in the present study. At station C0608, the effect from the Kuroshio mainstream may have further resulted in an increase in primary productivity and OM input, as reflected in the increased δ¹³C values, and THAA, NS, TOC, and TN concentrations, but the OM underwent severe degradation due to the depth of the water column. Furthermore, PCA results indicated that distance and depth have a positive effect at stations
C0508 and C0608. Moreover, the low Λ8 values with the highest OC content at C0608 may be associated with the dilution of $OC_{terr}$ by $OC_{marine}$. Past work has revealed that there is near-bottom lateral transport of suspended sediments from the ECS inner shelf to the trough, such that terrestrial OM at C0508 and C0608 may have been mainly derived from cross-shelf transport (Mei et al., 2019). Indeed, based on characterization of the terrestrial vegetation, C0508 and C0608 contained angiosperms, in contrast to the plant types occurring in the inner ECS. The transport of terrestrial OM from the coast and
nearby islands towards the KC and OT by typhoons and currents cannot be ignored (Bian et al., 2010; Chen et al., 2017). Prior research demonstrated that some terrestrial OM is transported southward along the Chinese coast and passes along the north of Taiwan. Subsequently these terrestrial OM sources join the northwardly flowing KC without traversing the central ECS (Chen et al., 2017). Moreover, some terrestrial material settles in the southern region of the OT by gravity, while fine particles are transported to the northern trough and OT by the KC (Mei et al., 2019). Yellow Sea mixed water may also
contribute terrestrial OM and can reach offshore regions near 30°N (Zhou et al., 2018b). Nevertheless, our findings indicated that the OM input by the KC had a major impact on the OT (C0608). However, due to the lack of direct evidence on nutrient and total chlorophyll pigment concentrations (CHL), as well as lack of data on bacteria and zooplankton, this inference needs to be confirmed by further studies.

    Woody gymnosperms, nonwoody gymnosperms, woody angiosperms and nonwoody angiosperms in prior studies (Tareq
et al., 2011; Zhu et al., 2011; Zhu et al., 2008) yielded LPVI values of 1, 3 to 27, 67 to 415, and 176 to 2782, respectively. The LPVI values in the present study indicated that terrestrial OM in the prodelta SOM, the shelf (at DHa-4), the offshore outer shelf (at DHa-5 and C0508) and the OT (C0608) were derived from nonwoody gymnosperms, woody angiosperms, nonwoody angiosperms and nonwoody angiosperms, respectively. However, the type source of terrestrial OM in the nearshore outer shelf (at DHa-2 and DHa-3) could not be determined by LPVI characterization. The C/V ratio serves as a
proxy for distinguishing woody (C/V < 0.1) and nonwoody (C/V > 0.1) plant tissue contributions (Tareq et al., 2011). In the present study C/V values increased offshore and exceeded 0.1 across stations DHa-2 to C0608, as shown in Figure 3, indicating that the nonwoody (leaf and grass) tissue fraction increased and nonwoody plant inputs dominated over woody vegetation sources, which was attributed to preferential transport and preservation of nonwoody vascular plant tissue to distal sediments. Lignin phenol partitioning occurred during cross-shelf transport. Therefore, C/V ratios indicated that SOM
at the shelf (DHa-2 and DHa-3) was derived from nonwoody tissue. However, results of shelf (DHa-4) S/V and C/V





analyses conducted in the present study reflected nonwoody gymnosperms/angiosperms yielding contradictory results with those of LPVI. Uncertainty in the identification of plant type using C/V and S/V ratios may be caused by the differential degradation rates of the monomers (C, V and S) during transport and burial, and may be circumvented using LPVI. Finally, woody tissue may be correlated with coarse minerals at DHa-4 as the sand content and median grain size were the highest at
this site which was dominated by woody plants. In addition, vegetation types indicated by S/V and C/V at the shelf (DHa-5 and C0508) and the OT (C0608) were consistent with LPVI. In summary,More nonwoody inputs from gymnosperms were found nearshore, and nonwoody inputs from angiosperms were observed offshore based on C/V, S/V, and LPVI values.

### 4.2 Reactivity of sedimentary organic matter as indicated by multiple biomarkers

Degradation of SOM is usually affected by the origin of SOM as well as sediment grain size and bacterial biomass (Dang et
al., 2014). These factors also alter SOM composition and abundance of biomarkers. Therefore, the application of a single marker to characterize SOM activity can bias the values of these parameters. To overcome the limitations of mono-proxy methods, OM relative diagenetic state was evaluated using three kinds of molecular indicators (lignin, NS, and THAA) in our study. Both the variation in P/(V+S) and (Ad/Al)v indicated that the degree of lignin demethylation and/or demethoxylation increased with increasing offshore distance. The high (Ad/Al)v and P/(V+S) values of surface sediments
suggest that terrestrial OM was already highly degraded. In the present study (Ad/Al)v was higher in upwelled mud along the main transport trajectory than along the coast (C0508 = 0.81 and C0501 = 0.45), indicating the higher oxidative state of terrestrial OM (Liu et al., 2021). Compared with (Ad/Al)v, (Ad/Al)s was not a good indicator (Fig. 3d).

The DI$_{AA}$ index decreased markedly, and the percentage of Gly increased (Fig. 5(b)) while that of Glc decreased (Fig. 4(b)) from the CJE to the shelf area with increasing water depth, reflecting severe diagenetic alteration offshore. This was
consistent with the degradation status indicated by lignin. However, it was intriguing that DI$_{AA}$ values appeared to be inconsistent with those of THAA(%TOC) along this transect. This is likely because each index is most sensitive to a particular diagenetic stage. DI$_{AA}$ was most effective in reflecting diagenetic alterations during intermediate stages of decomposition over timescales of years to decades, whereas THAA(%TOC) and THAA(%TN) were more sensitive indicators of early stages of OM degradation stages (Chen et al., 2018; Davis et al., 2009). Lignin values of (Ad/Al)v and
P/(V+S) reflected the diagenetic state of terrestrial OM. In summary, when SOM was mainly terrigenous, the assessment of SOM reactivity could be based on lignin indicators. In turn, when marine sources were dominant, SOM reactivity was reflected by the THAA(%TOC).

Values of THAA(%TOC) and NS(%TOC) were lower at C0501 than at CFJA in the prodelta, in agreement with SR values. Owing to their dominance in the turbid zone of the estuary (Table S2), OM particles were likely repeatedly
resuspended and repartitioned resulting in long particle residence times, thus making the residual THAA and NS relatively refractory. Additionally, the input of $OC_{terr}$ diluted $OC_{marine}$ inputs at station C0501 (Table S1). A relatively low C/N ratio and high THAA (Fig. 5) and Gal content (Fig. 4), and high THAA(%TOC) were indicative of the higher productivity and



fresher OM at the farthest outer shelf (C0508) (Vijayan et al., 2023). Particle size and composition at station C0508 was the same as that at OT station C0608; however, THAA(%TOC), THAA(%TN), and NS(%TOC) values were lower at C0608,

suggesting that a "grain-size effect" (Liu et al., 2021; Tareq et al., 2004; Zhu et al., 2011) could not explain the OM degradation state at this station. The high C/N ratio and high $\delta^{15}N$ value at C0608 (5.4‰) comparable to the average $\delta^{15}N$ value (5.5 to 6.1‰) of nitrate in Kuroshio water (Liang et al., 2023; Wu et al., 2003), suggested the microbial assimilation of inorganic N into particulate OM and its input by the KC (Fig. S4). The effect of water depth at C0608 on OM degradation cannot be ignored. All molecular biomarkers were consistent with $F_{burial}$, indicating that OM at C0608 had undergone

extensive biodegradation. Thus, the critical role of hydrodynamic processes and a combination of biomarkers need to be considered in evaluating the fate of OM.

In this survey, the trends in NS(%TOC) and THAA(%TOC) values with distance from the estuary were similar, but the rate of change in NS(%TOC) with distance from shore was lower than that of THAA(%TOC). Such disparities between NS(%TOC) and THAA(%TOC) could result from their different production mechanism and function, as well as different

sensitivities to diagenesis and different relative content (Grutters et al., 2002; Lehmann et al., 2020; Zhu et al., 2011). Additionally, in the present study the THAA content was about twice that of NS, so the biodegradability of OM may also differ (Matiasek and Hernes, 2019). Results of the present study suggested that NS is helpful to further support THAA results.

## 4.3 Differential burial of sedimentary organic matter

Anthropogenically induced climate change has modified global ocean conditions, raising concerns about $OC_{marine}$ and $OC_{terr}$ in sedimentary OC and $F_{burial}$ (Song et al., 2022). The latter decreased seaward ranking as follows: prodelta sediments > shelf sediments > the OT. Continuous resuspension and redeposition cause regular oxidation–reduction cycles that speed up OM decomposition and enhance the remineralization of sedimentary OC, which can reduce $F_{burial}$ (Song et al., 2022). $F_{burial}$ values estimated for the deltaic region in the present study were ten times greater than the global value (2.6 g

$C \cdot m^{-2} \cdot yr^{-1}$) for similar marine environments, but were comparable to those averaged for the delta of the Amazon River (Sobrinho et al., 2021). Average $F_{burial}$ along the transect in this study was estimated to be 16.7 g $C \cdot m^{-2} \cdot yr^{-1}$, higher than that of the Yellow Sea (15.3 g·$C \cdot m^{-2} \cdot yr^{-1}$) (Liao et al., 2018). The >70% $f_{marine}$ value of buried OC in the shelf and the OT in modern times indicated that marine productivity plays an important role in OC burial. Thus, $OC_{marine}$ was mainly photosynthetically derived, and buried OC in the transect may serve as a sink for atmospheric carbon dioxide ($CO_2$). As

global warming intensifies, enhanced stratification driven by the elevated sea surface temperature (SST) slows down upwelling of seafloor nutrients. Thus, CHL levels, as an indicator of phytoplankton production and biomass, have declined (Boyce et al., 2010; Venegas et al., 2025). At the same time, the remineralization rate of OC may increase with global warming (Guo et al., 2021b). These factors will lead to a marked decrease in the amount of carbon in shelf sediments.



## 5. Conclusions

The spatio-temporal variation in SOM in this study was characterized by three molecular biomarkers, and bulk and stable isotope values along a transect from the CJE to the OT. Marine OM was the main source of SOM along the prodelta-OT transect, whereas OM in nearshore areas was greatly influenced by terrestrial inputs. Terrestrial vegetation debris along the transect was mainly derived from nonwoody plants, with increasing oxidation, demethylation, and degradation by demethoxylization observed in a seaward direction. The THAA(%TOC) index was more sensitive than NS(%TOC) for

detecting diagenesis. Compared with $DI_{AA}$, THAA(%TOC), THAA(%TN), and NS(%TOC) were more sensitive indicators for the detection of the initial degradation stage of OM. Additionally, the quantity and quality of SOM parameters did not show consistent changes offshore, which may have resulted due to hydrological conditions and sedimentary types in this region. The nearest to shore outer shelf station (DHa-2) was markedly affected by the NKBC, whereas the farthest outer shelf station (C0508) and OT station C0608 were affected mainly by the KC rather than the CDW. Our results indicated that

the composition and degradation state of SOM was markedly affected by primary production, hydrodynamics, and sediment type. $F_{burial}$ along the study transect was dependent on the SR, except at the prodeltaic station (C0501). $OC_{terr-plant}$ and $OC_{plant}$ were mainly deposited in the prodelta and OT. Burial of predominantly $OC_{marine}$ components ($f_{marine} > 52.3\%$) implies that the transect functions as a carbon sink in the global carbon cycle. Our data reinforce the fact that the KC and CDW, which carry sediments and nutrients, influence the spread of OM in surface sediments, thus improving our

understanding of biogeochemical processes. Our study further demonstrates the importance of using a multiproxy approach to fully elucidate and quantify the relative contributions of OM from different sources along a sediment transect.

### Acknowledgments

The authors gratefully acknowledge colleagues from the Laboratory of Marine Chemistry Theory and Technology, Ocean University of China, and the crew of R/V *Dong Fang Hong* 2 and *Science 3* for their assistance with sampling. This study

was funded by the Shandong Key Research and Development Program (grant No. 2022QNLM040002), and Fundamental Research Funds for the Central Universities (grant No. 202464007, 202472007).

### Competing Interests

The authors declare that they have no conflict of interest.

### Data availability

Data will be made available upon request.





**Supplement**

Supplementary material to this article can be found online at https://data.mendeley.com/preview/v5p66bvy6j?a=39ed4c7b-dab3-4e1e-b15d-7ae5f7ea2ca1

**Author contributions**

SL (Shengkang Liang) and XY were responsible for conceptualization and review and editing of the manuscript. XY, SL (Shengkang Liang) and GZ performed the data analysis. SL (Shengkang Liang), GZ, XY, SL (Shanshan Li), HH and HM planned the methodology. XY was responsible for writing of the original draft. SL (Shengkang Liang) and GZ contributed the resources, and SL (Shengkang Liang) was responsible for supervision, funding acquisition and project administration. GZ conducted the research.

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
