# Peer review of "Sources, Reactivity and Burial of Organic Matter in East China Sea Sediments, as Indicated by a Multi-geochemical Proxy Approach"

_EGUsphere, 2025_

## Author Comment (AC1)

**Response to reviewers of the manuscript entitled "Sources, Reactivity and Burial of Organic Matter in East China Sea Sediments, as Indicated by a Multi-geochemical Proxy Approach" authored by X. Yu, S. Liang, G. Zhang, S. Li, H. Huang and H. Ma.**

Below, the reviewer comments are included in blue, and responses in black font. All page/line numbers referenced below refer to the preprint.

**Reviewer #1**

The authors measured a suite of biogeochemical proxies including multi-biomarkers in surface sediments from an East China Sea shelf transect to investigate the sources and reactivity of OM. This paper provides valuable insights into OM transformation and carbon cycling across the land-ocean interface. With the benefit of multiproxy, terrestrial and marine derived OM, vegetation sources and degradation conditions were tried to be examined. Overall, this paper provides a valuable package of data with insights into OM transformation and carbon cycling across the land-ocean interface. While I support this manuscript for publication after a major revision, I do have some comments/concerns they need to address.

**Major comments**

**Comment 1.(1)** The authors mention that this transect provides an ideal area to investigate the hydrodynamic-driven redistribution of OC. However, the manuscript lacks detailed descriptions of the hydrodynamic conditions. A more comprehensive introduction and explanation of how hydrodynamic data were measured or utilized would strengthen the manuscript.

**Response:** Our study area covers the estuary, continental shelf, and the Okinawa Trough, and is influenced by a variety of hydrodynamic processes, including the Changjiang Diluted Water, the nearshore and offshore Kuroshio Branch Currents, and the Taiwan Warm Current. These conditions make it an ideal region for investigating the hydrodynamic-driven redistribution of OC. Numerous studies have conducted that included detailed observations and modeling of the region's hydrodynamic characteristics. In this study, we have relied on these published studies to obtain the relevant hydrodynamic information. We will add description of the hydrodynamic features and their influence on the study area in Line 120, Page 4, Part 2.1, as follows:

"The YSCC flows southward throughout the year but is unable to penetrate the ECS during summer. In winter, however, it extends deeply into the northern ECS (Hwang et al., 2014). Together with the Yellow Sea Warm Current (YSWC), the YSCC forms a circulation system that induces upwelling, thus transporting nutrient-rich waters from deeper layers to the surface and enhancing primary productivity (Wang et al., 2019). The CDW contributes significantly to the nutrient pool of the ECS. During summer, most river-borne sediments are temporarily deposited in the subaqueous delta and estuarine zones. In winter, CDW flows southwestward along the coast toward the Taiwan Strait, and, in conjunction with the southward-flowing ZFCC, facilitates the transport of resuspended sediments (Hwang et al., 2014; Hu et al., 2014). The TWC flows northward year-round along the 50–100 m isobaths. In summer, the TWC originates from warm water in the Taiwan Strait and Kuroshio subsurface water, whereas in winter, it is primarily sustained by the Kuroshio subsurface intrusion, which reaches as far north as the subaqueous valley off the Changjiang Estuary (CJE) (Lian et al., 2016). Both the OKBC and the nearshore Kuroshio Branch Current (NKBC) deliver nutrient-enriched waters, particularly in phosphorus (P), thereby promoting primary productivity (Yang et al., 2012; Yang et al., 2013). The OKBC, flowing at depths of 60–120 m, moves northwestward along the 100 m isobath and partially rejoins the Kuroshio Main Stream near 28°N. The NKBC, originating at depths of 120–250 m, turns upward at approximately 27.5°N, 122°E, and then flows northeastward along the 60 m isobath until it reaches 30.5°N before veering eastward (Yang et al., 2012; Yang et al., 2013) (Fig. 1)."

(2)Regarding Figure 1 and hydrodynamic description: Hydrodynamics and currents are also important components in this paper, especially in the ocean part. Even though this figure 1 is presented in 3-D, it does not properly show the 'spatial range (i.e., approximate depth and range) of key current (KC, TWC, CDW…)'. Additionally, the sampling stations are difficult to identify. I recommend revising Figure 1 to improve clarity and informational value.

**Response:** To improve the clarity of station locations and current patterns, we have redrawn Figure 1 using a 2D map format. In the revised Figure 1, we unified the color scheme for sampling stations and sediment types to enhance visual consistency and informational clarity. The updated Figure 1, shown below:

[Figure]

**Figure 1.** Map showing the location of sampling stations in the study transect and regional ocean circulation patterns in spring. KC, Kuroshio Current; TWC, Taiwan Warm Current; YSCC, Yellow Sea Coastal Current; ZFCC, Zhejiang-Fujian Coastal Current; CDW, Changjiang Diluted Water; OKBC, offshore Kuroshio Branch Current; NKBC, nearshore Kuroshio Branch Current; KC, Kuroshio Current; TSWC, Tsushima Warm Current; CHWC, Cheju Warm Current.

**Comment 2.** This paper includes a nice dataset of biogeochemical parameters, and I believe this could be beneficial for other researchers working on the East China Sea shelf region. I recommend the authors include a summary table (maybe in supplementary) listing the proxies used, their function or roles in this study, and relevant references. This would be particularly helpful for readers who are less familiar with these parameters.

**Response:** As suggested, we have included a summary table (Table S4) of the Supplementary Materials. The table lists the proxies used in this study, their functions or roles, and the relevant references. The detailed content is as follows:

**Table S4.** The summary of proxies and their functions in surface sediments in the East China Sea.

| Proxy | Function | Parameters | Values | Indication | References |
|---|---|---|---|---|---|
| C/N | Sources | the ratio of total organic carbon to total nitrogen (C/N) | >20 | terrestrial organic matter | (Ankit et al., 2022) |
| | | | 4 to 12 | marine organic matter | (Zhou, 2016; Tesi et al., 2007; Wang et al., 2021) |
| $\delta^{13}C$ | Sources | stable carbon isotope ($\delta^{13}C$) | -24‰ to -18‰ | marine phytoplankton | (Martens et al., 2019; Duan et al., 2019) |
| | | | -33‰ to -24‰ | terrestrial plants with the Calvin-Benson cycle (C3) | (Liu et al., 2020) |
| | | | -16‰ to -9‰ | terrestrial plants with the Hatch-Slack cycle (C4) | (Wang et al., 2018a) |
| | | | -20‰ to -10‰ | terrestrial plants with the Crassulacean acid metabolism cycle (CAM) | (Liu et al., 2020) |
| $\delta^{15}N$ | Sources | stable nitrogen isotope ($\delta^{15}N$) | 3‰ to 12‰ | marine organic matter; polluted rivers (>4‰); less polluted and large rivers (≈ 4‰) | (Wang et al., 2018a; Voss et al., 2006) |
| | | | ≈ 0‰ | terrestrial organic matter; organic matter derived from *Trichodesmium*; pristine mountain rivers (< 4‰) | (Gireeshkumar et al., 2013; Voss et al., 2006) |
| Lignin | Vegetable types | syringyl to vanillyl (S/V) | ≈ 0 | gymnosperms | (Tareq et al., 2004) |
| | | | 0.6 to 4 | angiosperms | (Tareq et al., 2011) |
| | | cinnamyl to vanillyl (C/V) | <0.1 | woody plants | (Tareq et al., 2011) |
| | | | >0.1 | nonwoody plants | |
| | | lignin phenol vegetation index (LPVI) | ≈1 | woody gymnosperms | (Zhu et al., 2008; Zhu et al., 2011; Tareq et al., 2011) |
| | | | 3 to 27 | nonwoody gymnosperms | |

| | | | | | |
|---|---|---|---|---|---|
| | | | 67 to 415 | woody angiosperm | |
| | | | 176 to 2782 | nonwoody angiosperm | |
| | Degradation state | acid to aldehyde ratios of vanillyl (Ad/Al)v and syringyl (Ad/Al)s | 0.1 to 0.3 | fresh plants | (Tareq et al., 2004; Zhou, 2016) |
| | | | >0.5 | highly degraded terrestrial organic matter | (Tareq et al., 2004; Goñi, 1997) |
| | | p-hydroxyl phenol to (vanillyl phenol + syringyl phenol) (P/(V+S)) | <0.39 | weakly degraded terrestrial organic matter | |
| | | | 0.39 to 0.63 | moderately degraded terrestrial organic matter | (Zhou, 2016) |
| | | | >0.63 | highly degraded terrestrial organic matter | |
| | Sources | aspartic acid to glycine (Asp/Gly); serine plus threonine (Ser+Thr) | Asp/Gly: diatomaceous materials ($\approx$0.62%) < calcareous materials ($\approx$1.88%). Ser+Thr: diatomaceous materials ($\approx$16.7%) > calcareous materials ($\approx$9%) | | (Müller et al., 1986; Wei et al., 2022) |
| THAA | | carbon-normalized amino acid yields (THAA(%TOC)) | | | |
| | Degradation state | nitrogen-normalized amino acid yields (THAA(%TN)) | decrease with increasing organic matter degradation | | (Wang et al., 2018b; Gaye et al., 2022) |
| | | degradation index (DI) | | | |
| NS | Degradation state | relative percentage of fucose and rhamnose to total neutral sugars (mol% (Rha+Fuc)) | increase with increasing organic matter degradation | | (He et al., 2010) |

| | | carbon-normalized neutral sugar yields (NS(%TOC)) | decrease with increasing organic matter degradation | (Lehmann et al., 2020; Amon et al., 2001; Amon and Benner, 2003) |
|---|---|---|---|---|

**Comment 3.** (1) Are there other possible terrestrial OM transport pathways in the study area (e.g., smaller rivers, submarine groundwater discharge, etc.)? A brief discussion of these factors would be helpful.

**Response:** In addition to the Changjiang River-dominated terrestrial OM input, other pathways also contribute to regional fluxes. We will add a brief discussion of these factors in Part 4.1 on Page 15, Line 335, as follows:

"Furthermore, native processes (e.g., submarine groundwater, monsoons, and smaller rivers), and human activities all contribute to the input of terrestrial OM (Sun et al., 2023; Wang et al., 2021). For example, significant pulsed discharges of high-salinity groundwater occur in the South Atlantic Bight, up to 15 km from the coast (George et al., 2020). Similarly, in the Zhejiang-Fujian coastal area of the East China Sea, results of CDOM (chromophoric dissolved organic matter) in surface seawater indicate that its source may likely be affected by groundwater input (Gao et al., 2010). Before 1970, the intensity of the East Asian winter monsoon was positively correlated with the amount of terrestrial OM burial in the muddy areas of the East China Sea (ECS) (Wang et al., 2021). In recent decades, sediment discharge has decreased by ≈70% mainly due to dam construction and other water/soil conservation measures (Wang et al., 2021)."

(2) Large river estuaries typically exhibit substantial seasonal variability. Could this variability introduce potential bias in the terrestrial OM input due to changes in both its quantity and composition?

**Response:** We thank the reviewer for raising the important issue of seasonal variability in river estuaries. To address this point, we will add a supplementary discussion in the manuscript in Part 4.1 on Page 15, at Line 352, as follows:

"Seasonal variation in large river estuaries can usually drive changes in the quantity and composition of terrestrial OM input to water bodies, greatly affecting the DOC and POC fluxes into the sea (Chen et al., 2025; Wang et al., 2012; Wang et al., 2023; Han et al., 2021). Previous studies in the east of Zhoushan Archipelago to the north of the Taiwan Strait have shown that the spring average percentage of terrigenous OM in surface sediments is higher than that in autumn (Zhou et al., 2018a). It is noteworthy that the sediment survey stations in the present study were all sampled during the same season, and the sampling time coincided with the CJR flood period (May-October), and is thus representative of the interannual variation and is minimally affected by seasonal

variability. In areas beyond the estuary to the Okinawa Trough, based on sedimentation rates, the age of surface sediments exceeds one year. Additionally, in anaerobic environments, the half-life of terrigenous biomarkers (such as lignin phenols) in sediments is at least 300 years (Dittmar and Lara, 2001), meaning their signals are not easily altered during seasonal variations."

(3) The sediment samples used in this study were collected in 2009 and 2010, which is approximately 15 years ago. Do the authors anticipate any bias due to this time gap, such as the impacts of human activities (e.g., dam construction, population growth, industrialization) or climate change over the intervening years?

**Response:** While we acknowledge that the sediment samples were collected in 2009 and 2010, i.e., approximately 15 years ago, the following evidence supports the continued scientific validity of our conclusions:

1. **sampling occurred after major human impacts had stabilized:** After the Three Gorges Dam began impoundment in 2003, the sediment discharge from the Yangtze River had already decreased by over 55% by 2009 (Chen et al., 2019). This year marked the full operational start of the Three Gorges Project in terms of impounding and regulating large-scale runoff. Given ongoing global warming and intensified human activities, our dataset serves as an important baseline. Our study thus provides a valuable reference for subsequent studies. Future sampling in the same region will allow more precise quantification of temporal changes.

2. **Hydrological stability:** the Yellow Sea Coastal Current, offshore Kuroshio Branch Current, and Changjiang Diluted Water studied here have been persistent over the past 15 years with limited variation in their pathways (Yang et al., 2012; Zhou et al., 2018b; Zhou et al., 2018a).

3. **Parameter stability:** Isotopes and lignin biomarkers are relatively stable and have long half-lives (Dittmar and Lara, 2001). Our calculations, together with comparisons to recent literature data, indicate no substantial bias due to the time gap.

4. **Relevance to study objectives:** Carbon burial in stable sedimentary environments occurs over long timescales (Song et al., 2022). Our study focuses on a multi-geochemical proxy approach and on carbon burial mechanisms rather than instantaneous conditions. Data from 2018, 2014, and 2012 show that the proportion of terrestrial OM in the ECS sediments still exhibits a seaward decreasing trend (Zhou et al., 2018a; Li et al., 2014; Walter et al., 2007), consistent with the spatial patterns observed in our study. Comparative analysis

of multiple proxies from the same samples in the present study, which are rare in the literature, provides valuable insights into OM sources, distribution, and burial.

We will add a supplementary discussion addressing these points in Part 4.1 on Page 14, Line 332 of the preprint, as follows:

"The spatial distribution patterns of OM in marine shelf sediments are mainly controlled by hydrodynamic sorting and nutrient concentrations, while their temporal variation is predominantly regulated by climate events and anthropogenic activities (Li et al., 2022). The seasonal durations of typhoons and strong winds during 2010–2019 increased, which may have increased the southward transport of POC like the northward passage of Typhoon Bolaven (Wang et al., 2023). Following impoundment of the Three Gorges Dam (TGD), the input of phytodetritus derived from freshwater phytoplankton increased, leading to a decreasing trend in $\delta^{13}C$ values of surface sediments along the Zhejiang-Fujian coast starting in 2006 (Yao et al., 2015). Regulation of the CJR runoff has resulted in weakening of the transport of terrigenous materials to the sea, and thereby in a marked reduction in the deposition of CJR terrestrial OM in the CJR Estuary and along the Zhejiang-Fujian coast (Li et al., 2022). Yang et al. (2021) confirmed that human activity has markedly reduced CJR sediment discharge, which in turn decreased the transport of nutrients and terrigenous materials. This process suppressed marine primary productivity, causing a decline in the burial flux of marine OC (Yang et al., 2021). Findings of our study are close to the data from Zhou et al. (2018a) and Li et al. (2014), demonstrating that the proportion of terrigenous OM in ECS sediments exhibits a seaward decreasing trend and the spatial pattern across temporal scales is stable."

**Comment 4.** It may not be straightforward, but is it possible for the authors to estimate the degradation or transport time of OM by combining geochemical evidence from the sediments with the known distances (in km) along the transect?

**Response:** Yes, it is feasible. We have estimated the degradation and transport time of OM and performed data fitting using power attenuation and first-order kinetics models, with transport time and distance from the estuary to each station as independent variables. These analyses will be incorporated into the revised manuscript, including the Abstract, Methods, Results, Discussion, and Conclusion sections. Detailed additions are as follows:

(1) **Abstract (Page 1, Line 21)**

The proxies of OM in the transect collectively fit a power-law model but with distinct attenuation rates (the a* values) for individual OC pool and biogeochemical indicators.

**(2) Materials and Methods(Part 2.5, Page 8, Line 212)**

2.5 Lateral transport time and degradation kinetic models

2.5.1 Lateral transport time

The age of organic carbon (kyr) and the transport time (kyr) were calculated on the base of Bao et al. (2016)'s large and accurate $^{14}C$ data dataset (Bao et al., 2019). The kriging interpolation method was applied to the target station. According to the formula proposed by Bao et al. (2019), further derivation processes are detailed in Text S1, yielding the following formula:

$$Age = 1/\lambda \ln \left(1/(1 + {}^{14}C/1000)\right) \qquad (12)$$

$$\Delta t = Age_B - Age_A \qquad (13)$$

where $\lambda$ is the "true" decay constant for radiocarbon: 0.000121 (Bao et al., 2019). Age (kyr) is the age of OC, and $\Delta t$ is the lateral transport time. A and B were the locations A and B, respectively.

According to Broder et al. (2018)'s method, the transport time (kyr) was also calculated as follows:

$$Age = a \times x + b \qquad (14)$$

$$\Delta t = Age - b \qquad (15)$$

where x (m) is the water depth. a and b defined the inverse net cross-shelf transport velocity and the pre-ageing on land, respectively.

2.5.2 Degradation kinetic models

The carbon pools were estimated from fitting different kinetic reaction function to the data:

A first-order decay model of SOM expressed as $C_t = C_0 \times e^{-k_1\Delta t}$ can be converted to (Chen et al., 2011):

$$ln\, C_t - \ln C_0 = -k_1\Delta t \qquad (16)$$

A power attenuation model is analogous to the first-order decay model which the degradability is subject to a power-law (Zhu et al., 2013). In this study, the distance from the river mouth (d) was substituted for the water depth as proposed by Zhu et al. (2013):

$$C_t = C_0 \times d^{-a^*} \qquad (17)$$

where $C_t$ (mg·g$^{-1}$sediment dry weight) is the concentration of a compound A at distance from the river mouth; $C_0$ (mg·g$^{-1}$sediment dry weight) is the initial concentration; a* is the compound-specific, distance-related attenuation constant, and d (km) is the distance from the river mouth. $k_1$ (kyr$^{-1}$) is the first-order degradation rate constant.

**(3) Results(Part 3.4, Page 14, Line 318)**

3.4 Cross-shelf transport time for sedimentary organic matter

The age of OC generally increases with increasing distance from the river mouth and water depth, whereas the OC age in the shelf sediments (C0508) and the OT sediments (C0608) is relatively young (Table 2). Thus, only data from C0501 to DHa-5 were subjected to linear fitting to get the lateral transport time (Fig. 7). The fitted parameters of Equation (14) were obtained as follows: a= 0.015±0.01 kyr·m$^{-1}$, b=2.79±0.24 kyr.

Three terrestrial OM components ($OC_{terr}$, $OC_{plant}$, and $OC_{terr-plant}$) and three biomarkers (lignin, THAA, and NS) were plotted against the distance from the river mouth and the transport time (Fig. 8 and Fig. 9). The R² values of different indices across various models indicate that when the distance from the river mouth is taken as the independent variable (power attenuation model), the R² values are higher than those when time is used as the independent variable (first-order decay model). The calculated attenuation constant a* values of different OC pool and biomarkers decrease in the order: $OC_{plant} > OC_{terr} > OC_{terr-plant}$, $\sum 8 > NS > THAA$, respectively (Fig. 8), with the higher a* values indicating quicker attenuation seaward. The variation pattern of k is consistent with that of a*. The transport time of other stations relative to C0501 calculated by the method proposed by Broder et al. (2018) (Equation 15) is consistent with the corresponding results obtained using the method put forward by Bao et al. (2019) (Equation 13).

**Table 2.** Radiocarbon data for bulk organic carbon and the lateral transport time for stations along the transect in the East Chins Sea

| Station | $\Delta^{14}C$ (‰) | Age (kyr) | Lateral transport time (kyr) |
|---------|--------|-----------|------------------------------|
| C0501 | -290.17 | 2.83 | 0.039 |
| CFJA | -351.02 | 3.57 | 0.78 |
| DHa-2 | -350.87 | 3.57 | 0.78 |
| DHa-3 | -360.82 | 3.70 | 0.91 |
| DHa-4 | -352.39 | 3.59 | 0.80 |
| DHa-5 | -355.73 | 3.63 | 0.84 |
| C0508 | -235.89 | 2.22 | - |
| C0608 | -251.20 | 2.39 | - |

Note. -: Unknown.

[Figure]

**Figure 7.** Calibrated radiocarbon ages of bulk organic carbon vs. water depth.

[Figure]

**Figure 8.** Proxy values in surface sediments plotted against the distance from the river mouth, describing power attenuation model at sampling stations along an East China Sea transect.

[Figure]

**Figure 9.** Proxy values in surface sediments plotted against the lateral transport time, describing first-order decay model at sampling stations along an East China Sea transect.

**(4) Discussion (Part 4.3, Page 18, Line 450)**

The cross-shelf transport time in this study should be understood as a net (unidirectional cross-shelf vector) transfer time and not the actual random-walk speed (Bao et al., 2019; Broder et al., 2018). The results from the power attenuation model and the first-order decay models indicate that distance from the river mouth is more suitable as an independent variable than lateral transport time for reflecting the degradation process. This may be attributed to factors such as sediment resuspension, ocean currents, and human activities, which complicate the relationship between lateral transport time, transport of sediments with different grain sizes, and terrestrial OM diffusion. THAA and NS were not applicable to these models, possibly due to their lability and potential for autochthonous production. In this study, the attenuation rate of terrestrial OM decreases with the distance from the river mouth in the ECS (Fig. 8). Thus, the assumption that time roughly equals distance from the river mouth in this transect is generally robust. Specifically, we propose that the distance from the river mouth can be used as a proxy for the entirety of diagenetic history in the ECS, including both lateral transport and vertical burial. Consequently, terrestrial OM deposited in the deep shelf has undergone both a longer transport time and a longer burial time, leading to a longer diagenetic history.

**(5) Conclusion (Part 5, Page 19, Line 479)**

The concentrations of terrestrial OM components at the transect can be best described

by power-law attenuation curves and the distance is more suitable than lateral transport time for reflecting the degradation.

**Text S1 Lateral transport time (Supplementary material)**

According to Bao et al. (2019), the age of organic carbon (kyr) and the transport time (kyr) were calculated on the base of Bao et al. (2016)'s large and accurate $^{14}$C data dataset. The kriging interpolation method was applied to the target station.

$$((^{14}C/^{12}C)_{sample}/(^{14}C/^{12}C)_{modern} - 1) \times 1000 = \delta^{14}C \tag{1}$$

$$R_{sample} = (^{14}C/^{12}C)_{sample} \tag{2}$$

$$R_{modern} = (^{14}C/^{12}C)_{modern} \tag{3}$$

$$F_m = R_{sample}/R_{modern} \tag{4}$$

The equation describing radioactive decay (t) is:

$$N_B = N_A e^{-\lambda t} \tag{5}$$

$$F_{m,B} = F_{m,A} e^{-\lambda t} \tag{6}$$

It can be derived from Equations (1-4) that $F_m = 1 + \Delta^{14}C/1000$ $\qquad$ (7)

The relationship can be written according to Equations (5-6),

$$N_A/N_B = F_{m,A}/F_{m,B} = (R_{sample,A}/R_{modern})/(R_{sample,B}/R_{modern}) \tag{8}$$

When A represents modern and B represents sample, Equations (8) could be expressed as:

$$N_A/N_B = 1/F_{m,B} \tag{9}$$

In our study, lateral transport time corresponds to radiocarbon decay occurring during the transport process; that is, t=Age (Bao et al., 2019). Combining Equations (5), and (9), the following equation can be obtained:

$$Age = 1/\lambda \ln (1/F_{m,B}) \tag{10}$$

Combining Equations (7) and (10), the following equation can be obtained:

$$Age = 1/\lambda \ln (1/(1 + {}^{14}C/1000)) \tag{11}$$

Next, the constraints from radioactive isotope ($^{14}$C) decay were utilized to derive transport time. The equation is:

$$\Delta t = Age_B - Age_A \tag{12}$$

where $N_A$ is the number of atoms of the radioactive isotope in sample A, and $N_B$ is the number of atoms left after time t. $\lambda$ is the "true" decay constant for radiocarbon: 0.000121, and $F_m$ is the fraction modern, where $R_{sample}$ is $^{14}C/^{12}C$ ratio for the sample

and $R_{modern}$ is $^{14}C/^{12}C$ ratio for isotopic fractionation-normalized standard in the year 1950 (Bao et al., 2019). Age (kyr) is the age of OC, where $F_{m,A}$, $F_{m,B}$ are Fm values of corresponding OM from locations A and B, respectively. $\Delta t$ is the lateral transport time.

**Minor comments**

**Comment 1.** Line 36: "whereas ~ in sediment" I feel this sentence suddenly popped up, but phosphorus is not a MAIN topic dealt in this paper. I recommend removing this part.

**Response:** Thank you for your comment. The original sentence mistakenly referred to phosphorus (P) instead of particulate organic carbon (POC), which was a typographical error by us. This has been corrected in the revised manuscript.

**Comment 2.** Line 46: sometimes "land-ocean" used but sometimes "land-sea". Please unify the vocabulary, unless the author wants to categorize something different (if so, please explain).

**Response:** We have unified the terminology and changed "land-sea" to "land-ocean" in Line 46 for consistency in the revised manuscript.

**Comment 3.** Line 50: Does C/N in this paper is OC/TN? Or TC? TN? Maybe it's better to clarify when it first appears. (I saw this information in line 146).

**Response:** Thank you for pointing this out. We have revised the manuscript to clarify that C/N refers to the ratio of TOC to TN. Definitions for C/N, TOC, and TN have been added when they first appear to ensure clarity for readers.

**Comment 4.** Line 71: a few ppm: in this ms, per mil (‰) kept used. I recommend sticking to the same unit.

**Response:** The unit was mistakenly changed during multiple rounds of our own revision. We have corrected it back to "a few ‰" to maintain consistency throughout the manuscript.

**Comment 5.** Line 125: 0-2 cm sediment samples were collected by which device? Boxcore? Multicore?

**Response:** Except for the DHa-2 station, which was sampled using a gravity corer, all other stations were sampled with a box corer.

We will supplement and improve the content in Line 125 of page 5 in part 2.1, which originally read: "A total of eight surface (0–2 cm) sediment samples were collected along the transect from the delta to the OT on board the R/V Dong Fang Hong 2 and

Science 3 scientific research vessels on May 2009 and June 2010 was sampled for analysis (Fig.1, Table 1, and Table S1)." to "Surface sediments were collected using a box corer on board the R/V Dong Fang Hong 2 and Science 3 scientific research vessels on May 2009 and June 2010, and the top 0-2 cm layer along the transect from the delta to the OT was sampled for analysis (Fig.1, Table 1, and Table S1). For the DHa-2 sample, a gravity corer was employed to obtain a sediment core, which was sectioned, and the 0-2 cm segment was selected as the surface sediment sample."

**Comment 6.** Line 143, 145: Inorganic carbon removing part is written twice.

**Response:** We have removed the redundant description of inorganic carbon removal in Line 145.

**Comment 7.** Line 170-174: Does Chl-a were measured with a YSI sonde and from the filter? Please clarify which method you used in this paper.

**Response:** Two methods were used to measure Chl-*a* in this study. During field investigations, Chl-*a* was measured in situ using a YSI sonde. The discussion regarding the higher Chl-*a* concentration in the mid-layer of station DHa-2 compared to the corresponding layer at station CFJA refers to the YSI results. We have clarified this in the Discussion section (Part 4.1, Page 15, Line 359) by adding "(YSI data)" after "the Chl-*a* concentration". In addition, we have corrected "The filtrate was used for Chl-*a* and nutrients analysis." (Part 2.2, Page 6, Line 174) in the revised manuscript.

**Comment 8.** Line 177 and Figure 6: N/C ratio? or C/N ratio?

**Response:** The ratio in Line 177 and Figure 6 originally referred to N/C. To avoid confusion, we have carefully revised the manuscript and standardized all relevant instances to the commonly used format "C/N".

**Comment 9.** Line 220: 16.5 +_0.5 (n=##) please provide the number of samples used for this calculation.

**Response:** The number of samples used for this calculation is 8. We have now added "(n=8)" in Line 220.

**Comment 10.** Line 249: Section 3.2, description of stable isotopic values is too short. You can write down more, or I recommend considering this section within section 3.1., as your stable isotopic values were also measured in bulk.

**Response:** Thanks for your valuable suggestion. We will merge the original Section 3.2 into Section 3.1, and it will be presented as Subsection 3.1.3. Accordingly, the original

Section 3.3 will be renumbered as Section 3.2, and Section 3.4 will be renumbered as Section 3.3.

**Figures and Tables**

**Comment 11.** Fig.2 and 3- please provide error bars-

**Response:** We acknowledge the importance of representing measurement variability and regret that we are unable to provide sample-specific error bars (e.g., standard deviations based on sample replicates) for the data points at each station in the figures. This limitation stems from the fact that duplicate analyses were not performed on individual samples during the original survey. Practical constraints, including limited sample quantity available from each station, the considerable analytical time requirements for the comprehensive suite of parameters measured, and associated analytical costs, precluded obtaining replicate measurements for each sample. Additionally, previous studies component indices typically were measured only once. This may be due to the fact that during the sample-testing process, the precision and accuracy can be ensured by determining standard materials or adding standard materials, and the stability of the instrument can be checked. Meanwhile, it is also affected by factors such as the amount of sample obtained, the cost of sample pretreatment, as well as the testing duration and cost.

Nevertheless, in the present study stringent quality control (QC) procedures were rigorously implemented throughout the analysis for all parameters to ensure data accuracy and precision:

(1) Stable Isotopes ($\delta^{13}C$, $\delta^{15}N$):

The certified isotopic standard BR2151 ($\delta^{13}C$ = -26.27 ± 0.15 ‰, $\delta^{15}N$ = 4.42 ± 0.29 ‰) was analyzed repeatedly (once every 5-8 samples; n = 6 total replicates) interspersed with the samples. The relative standard deviations (RSD%) calculated from these replicates were <1.0‰ for $\delta^{13}C$ and 5.0% for $\delta^{15}N$ (Part 2.2, Page 6, Line 151).

(2) Lignin, neutral sugars (NS), total hydrolyzable amino acids (THAA):

For the analysis of them, we employed internal standards and calculated their recovery rates to monitor and ensure both accuracy and precision. Specifically, for NS, the RSD it is now explicitly stated that calculation was based on reference material measurements (Part 2.2, page 6, line 169). Our team routinely implements well-established methods for amino acid analysis, as indicated by peer-reviewed publications (Liang et al., 2023b; Liang et al., 2023a; Guo et al., 2021).

(3) TOC and TN: Data for the stations used in this study represent final values derived from results of a previously published, large-scale survey (Zhang et al., 2014). If

deemed essential, however, we can attempt to re-analyze archived samples (there should still be some remaining lyophilized material stored in the refrigerator).

**Comment 12.** Fig. S1-I recommend drawing a stacked-bar graph, instead of this.

**Response:** We have revised Figure S1 as suggested, replacing it with a stacked-bar graph. The revised figure is shown below:

[Figure]

**Figure S1.** Spatial distribution of mean particle size and percent composition in surface sediments of the East China Sea study transect.

**Comment 13.** Overall graphs-Some minor ticks represent a number with decimal places. Please adjust the number of minor ticks so it represents integers.

**Response:** We have adjusted the number of minor ticks in the revised manuscript so it represents integers. For example, the minor ticks in Figures S1, S2, S3, and S4 have been adjusted to display integer values, and the figures have been redrawn accordingly.

**Reference**

Amon, R. M. W. and Benner, R.: Combined neutral sugars as indicators of the diagenetic state of dissolved organic matter in the Arctic Ocean, Deep-Sea Res. Part I-Oceanogr. Res. Pap., 50, 151-169, https://doi.org/10.1016/S0967-0637(02)00130-9, 2003.

Amon, R. M. W., Fitznar, H. P., and Benner, R.: Linkages among the bioreactivity, chemical composition, and diagenetic state of marine dissolved organic matter, Limnol. Oceanogr., 46, 287-297, https://doi.org/10.4319/lo.2001.46.2.0287, 2001.

Ankit, Y., Muneer, W., Gaye, B., Lahajnar, N., Bhattacharya, S., Bulbul, M., Jehangir, A., Anoop, A., and MIshra, P. K.: Apportioning sedimentary organic matter sources and its degradation state: Inferences based on aliphatic hydrocarbons, amino acids and delta N-15, Environ. Res., 205, 112409, https://doi.org/10.1016/j.envres.2021.112409, 2022.

Bao, R., McIntyre, C., Zhao, M., Zhu, C., Kao, S.-J., and Eglinton, T. I.: Widespread dispersal and aging of organic carbon in shallow marginal seas, Geology, 44, 791-794, https://doi.org/10.1130/g37948.1, 2016.

Bao, R., Zhao, M., McNichol, A., Galy, V., McIntyre, C., Haghipour, N., and Eglinton, T. I.: Temporal constraints on lateral organic matter transport along a coastal mud belt, Org. Geochem., 128, 86-93, https://doi.org/10.1016/j.orggeochem.2019.01.007, 2019.

Broder, L., Tesi, T., Andersson, A., Semiletov, I., and Gustafsson, O.: Bounding cross-shelf transport time and degradation in Siberian-Arctic land-ocean carbon transfer, Nat. Commun., 9, https://doi.org/10.1038/s41467-018-03192-1, 2018.

Chen, D., Liu, Q., Xu, J., and Wang, K.: Model-Based Evaluation of Hydroelectric Dam's Impact on the Seasonal Variabilities of POC in Coastal Ocean: A Case Study of Three Gorges Project, J. Mar. Sci. Eng., 7, https://doi.org/10.3390/jmse7090320, 2019.

Chen, J. L., Wong, M. H., Wong, Y. S., and Tam, N. F. Y.: Modeling sorption and biodegradation of phenanthrene in mangrove sediment slurry, J. Hazard. Mater., 190, 409-415, https://doi.org/10.1016/j.jhazmat.2011.03.060, 2011.

Chen, S., Lou, S., Yang, Z., Zhang, Z., Liu, S., and Fedorova, I. V.: Tidal dynamics and seasonal hydrological variations influencing organic carbon distribution in the Yangtze River estuary, Mar. Environ. Res., 207, 107057, https://doi.org/10.1016/j.marenvres.2025.107057, 2025.

Dittmar, T. and Lara, R. J.: Molecular evidence for lignin degradation in sulfate-reducing mangrove sediments (Amazonia, Brazil), Geochim. Cosmochim. Acta, 65, 1417-1428, https://doi.org/10.1016/s0016-7037(00)00619-0, 2001.

Duan, L. Q., Song, J. M., Yuan, H. M., Li, X. G., and Peng, Q. C.: Occurrence and origins of biomarker aliphatic hydrocarbons and their indications in surface sediments of the East China Sea, Ecotoxicol. Environ. Saf., 167, 259-268, https://doi.org/10.1016/j.ecoenv.2018.10.011, 2019.

Gao, L., Fan, D., Li, D., and Cai, J.: Fluorescence characteristics of chromophoric dissolved organic matter in shallow water along the Zhejiang coasts, southeast China, Mar. Environ. Res., 69, 187-197, https://doi.org/10.1016/j.marenvres.2009.10.004, 2010.

Gaye, B., Lahajnar, N., Harms, N., Paul, S. A. L., Rixen, T., and Emeis, K.-C.: What can we learn from amino acids about oceanic organic matter cycling and degradation?, Biogeosci., 19, 807-830, https://doi.org/10.5194/bg-19-807-2022, 2022.

George, C., Moore, W. S., White, S. M., Smoak, E., Joye, S. B., Leier, A., and Wilson, A. M.: A New Mechanism for Submarine Groundwater Discharge From Continental Shelves, Water Resour. Res., 56, e2019WR026866, https://doi.org/10.1029/2019WR026866, 2020.

Gireeshkumar, T. R., Deepulal, P. M., and Chandramohanakumar, N.: Distribution and sources of sedimentary organic matter in a tropical estuary, south west coast of India (Cochin estuary): A baseline study, Mar. Pollut. Bull., 66, 239-245, https://doi.org/10.1016/j.marpolbul.2012.10.002, 2013.

Goñi, M. A.: Record of terrestrial organic matter composition in Amazon fan sediments, Proceedings of the Ocean Drilling Program, Scientific Results, 155, 519-530, https://doi.org/10.2973/odp.proc.sr.155.240.1997, 1997.

Guo, J. Q., Yuan, H. M., Song, J. M., Li, X. G., Duan, L. Q., Li, N., and Wang, Y. X.: Evaluation of Sedimentary Organic Carbon Reactivity and Burial in the Eastern China Marginal Seas, J. Geophys. Res.-Oceans, 126, e2021JC017207, https://doi.org/10.1029/2021JC017207, 2021.

Han, L., Wang, Y., Xiao, W., Wu, J., Guo, L., Wang, Y., Ge, H., and Xu, Y.: Seasonal Changes of Organic Carbon Mixing, Degradation and Deposition in Yangtze River Dominated Margin Related to Intrinsic Molecular and External Environmental Factors, J. Geophys. Res.-Biogeosci., 126, https://doi.org/10.1029/2021jg006637, 2021.

He, B., Dai, M., Huang, W., Liu, Q., Chen, H., and Xu, L.: Sources and accumulation of organic carbon in the Pearl River Estuary surface sediment as indicated by elemental, stable carbon isotopic, and carbohydrate compositions, Biogeosci., 7, 3343-3362, https://doi.org/10.5194/bg-7-3343-2010, 2010.

Hu, B., Li, J., Zhao, J., Wei, H., Yin, X., Li, G., Liu, Y., Sun, Z., Zou, L., Bai, F., Dou,

Y., Wang, L., and Sun, R.: Late Holocene elemental and isotopic carbon and nitrogen records from the East China Sea inner shelf: Implications for monsoon and upwelling, Mar. Chem., 162, 60-70, https://doi.org/10.1016/j.marchem.2014.03.008, 2014.

Hwang, J. H., Van, S. P., Choi, B.-J., Chang, Y. S., and Kim, Y. H.: The physical processes in the Yellow Sea, Ocean Coastal Manage., 102, 449-457, https://doi.org/10.1016/j.ocecoaman.2014.03.026, 2014.

Lehmann, M. F., Carstens, D., Deek, A., McCarthy, M., Schubert, C. J., and Zopfi, J.: Amino acid and amino sugar compositional changes during in vitro degradation of algal organic matter indicate rapid bacterial re-synthesis, Geochim. Cosmochim. Acta, 283, 67-84, https://doi.org/10.1016/j.gca.2020.05.025, 2020.

Li, D., Yao, P., Bianchi, T. S., Zhang, T., Zhao, B., Pan, H., Wang, J., and Yu, Z.: Organic carbon cycling in sediments of the Changjiang Estuary and adjacent shelf: Implication for the influence of Three Gorges Dam, J. Mar. Syst., 139, 409-419, https://doi.org/10.1016/j.jmarsys.2014.08.009, 2014.

Li, Y., Lin, J., Xu, X. P., Liu, J. Z., Zhou, Q. Z., and Wang, J. H.: Multiple biomarkers for indicating changes of the organic matter source over the last decades in the Min-Zhe sediment zone, the East China Sea, Ecol. Indic., 139, https://doi.org/10.1016/j.ecolind.2022.108917, 2022.

Lian, E., Yang, S., Wu, H., Yang, C., Li, C., and Liu, J. T.: Kuroshio subsurface water feeds the wintertime Taiwan Warm Current on the inner East China Sea shelf, J. Geophys. Res.-Oceans, 121, 4790-4803, https://doi.org/10.1002/2016JC011869, 2016.

Liang, S., Zhang, M., Wang, X., Li, H., Li, S., Ma, H., Wang, X., and Rong, Z.: Seasonal dynamics of dissolved organic matter bioavailability coupling with water mass circulation in the South Yellow Sea, Sci. Total Environ., 904, 166671, https://doi.org/10.1016/j.scitotenv.2023.166671, 2023a.

Liang, S. K., Li, S. S., Guo, J. Q., Yang, Y. Q., Xu, Z. H., Zhang, M. Z., Li, H. G., Yu, X. H., Yang, M. H., and Wang, X. L.: Source, composition, and reactivity of particulate organic matter along the Changjiang Estuary salinity gradient and adjacent sea, Mar. Chem., 252, 104245, https://doi.org/10.1016/j.marchem.2023.104245, 2023b.

Liu, C., Li, Z. W., Berhe, A. A., and Hu, B. X.: The isotopes and biomarker approaches for identifying eroded organic matter sources in sediments: A review, Adv. Agron., 162, 257-303, https://doi.org/10.1016/bs.agron.2020.02.005, 2020.

Martens, J., Wild, B., Pearce, C., Tesi, T., Andersson, A., Broder, L., O'Regan, M., Jakobsson, M., Skold, M., Gemery, L., Cronin, T. M., Semiletov, I., Dudarev, O.

V., and Gustafsson, O.: Remobilization of Old Permafrost Carbon to Chukchi Sea Sediments During the End of the Last Deglaciation, Global Biogeochem. Cycles, 33, 2-14, https://doi.org/10.1029/2018gb005969, 2019.

Müller, P. J., Suess, E., and AndréUngerer, C.: Amino acids and amino sugars of surface particulate and sediment trap material from waters of the Scotia Sea, Deep-Sea Res. Part I-Oceanogr. Res. Pap., 33, 819-838, https://doi.org/10.1016/0198-0149(86)90090-7 1986.

Song, S. S., Santos, I. R., Yu, H. M., Wang, F. M., Burnett, W. C., Bianchi, T. S., Dong, J. Y., and ...... A global assessment of the mixed layer in coastal sediments and implications for carbon storage, Nat. Commun., 13, 4903, https://doi.org/10.1038/s41467-022-32650-0, 2022.

Sun, Y., Wang, G., Weng, Y., Li, Q., Zhang, F., Jiang, W., Dai, G., Lin, W., Sun, S., Jiang, Y., and Zhang, Y.: Submarine groundwater discharge in Dongshan Bay, China: A master regulator of nutrients in spring and potential national significance of small bays, Front. Mar. Sci., 10, https://doi.org/10.3389/fmars.2023.1164589, 2023.

Tareq, S. M., Kitagawa, H., and Ohta, K.: Lignin biomarker and isotopic records of paleovegetation and climate changes from Lake Erhai, southwest China, since 18.5kaBP, Quat. Int., 229, 47-56, https://doi.org/10.1016/j.quaint.2010.04.014, 2011.

Tareq, S. M., Tanaka, N., and Ohta, K.: Biomarker signature in tropical wetland: lignin phenol vegetation index (LPVI) and its implications for reconstructing the paleoenvironment, Sci. Total Environ., 324, 91-103, https://doi.org/10.1016/j.scitotenv.2003.10.020, 2004.

Tesi, T., Miserocchi, S., Goñi, M. A., and Langone, L.: Source, transport and fate of terrestrial organic carbon on the western Mediterranean Sea, Gulf of Lions, France, Mar. Chem., 105, 101-117, https://doi.org/10.1016/j.marchem.2007.01.005, 2007.

Voss, M., Deutsch, B., Elmgren, R., Humborg, C., Kuuppo, P., Pastuszak, M., Rolff, C., and Schulte, U.: Source identification of nitrate by means of isotopic tracers in the Baltic Sea catchments, Biogeosci., 3, 663-676, https://doi.org/10.5194/bg-3-663-2006, 2006.

Walter, L. M., Ku, T. C. W., Muehlenbachs, K., Patterson, W. P., and Bonnell, L.: Controls on the δ13C of dissolved inorganic carbon in marine pore waters: An integrated case study of isotope exchange during syndepositional recrystallization of biogenic carbonate sediments (South Florida Platform, USA), Deep-Sea Res. Part I-Oceanogr. Res. Pap., 54, 1163-1200, https://doi.org/10.1016/j.dsr2.2007.04.014, 2007.

Wang, C., Lv, Y., and Li, Y.: Riverine input of organic carbon and nitrogen in water-sediment system from the Yellow River estuary reach to the coastal zone of Bohai Sea, China, Cont. Shelf Res., 157, 1-9, https://doi.org/10.1016/j.csr.2018.02.004, 2018a.

Wang, H. W., Kandasamy, S., Liu, Q. Q., Lin, B. Z., Lou, J.-Y., Veeran, Y., Huaiyan, L., Zhifei, L., and Arthur, C. C.-T.: Roles of sediment supply, geochemical composition and monsoon on organic matter burial along the longitudinal mud belt in the East China Sea in modern times, Geochim. Cosmochim. Acta, 305, 66-86, https://doi.org/10.1016/j.gca.2021.04.025, 2021.

Wang, K., Chen, J. F., Jin, H. Y., Li, H. L., and Zhang, W. Y.: Organic matter degradation in surface sediments of the Changjiang estuary: Evidence from amino acids, Sci. Total Environ., 637-638, 1004-1013, https://doi.org/10.1016/j.scitotenv.2018.04.242, 2018b.

Wang, X., Ma, H., Li, R., Song, Z., and Wu, J.: Seasonal fluxes and source variation of organic carbon transported by two major Chinese Rivers: The Yellow River and Changjiang (Yangtze) River, Global Biogeochem. Cycles, 26, https://doi.org/10.1029/2011gb004130, 2012.

Wang, Z., Bai, Y., He, X., Wu, H., Bai, R., Li, T., Zhu, B., and Gong, F.: Assessing the effect of strong wind events on the transport of particulate organic carbon in the Changjiang River estuary over the last 40 years, Remote Sens. Environ., 288, https://doi.org/10.1016/j.rse.2023.113477, 2023.

Wang, Z., Xiao, X., Yuan, Z., Wang, F., Xing, L., Gong, X., Kubota, Y., Uchida, M., and Zhao, M.: Air-sea interactive forcing on phytoplankton productivity and community structure changes in the East China Sea during the Holocene, Glob. Planet. Change, 179, 80-91, https://doi.org/10.1016/j.gloplacha.2019.05.008, 2019.

Wei, J. E., Chen, Y., Zhang, N., Yang, J. Q., Chen, R., Zhang, H. H., and Yang, G. P.: Variability and composition of amino acids and amino sugars in sediment cores of the Changjiang Estuary, Org. Geochem., 163, 104330, https://doi.org/10.1016/j.orggeochem.2021.104330, 2022.

Yang, D., Yin, B., Sun, J., and Zhang, Y.: Numerical study on the origins and the forcing mechanism of the phosphate in upwelling areas off the coast of Zhejiang province, China in summer, J. Mar. Syst., 123, 1-18, https://doi.org/10.1016/j.jmarsys.2013.04.002, 2013.

Yang, D. Z., Yin, B. S., Liu, Z. L., Bai, T., Qi, J. F., and Chen, H. Y.: Numerical study on the pattern and origins of Kuroshio branches in the bottom water of southern East China Sea in summer, J. Geophys. Res.-Oceans, 117, c02014,

https://doi.org/10.1029/2011jc007528, 2012.

Yang, Q., Qu, K., Yang, S., Sun, Y., Zhang, Y., and Zhou, M.: Environmental factors affecting regional differences and decadal variations in the buried flux of marine organic carbon in eastern shelf sea areas of China, Acta Oceanol. Sin., 40, 26-34, https://doi.org/10.1007/s13131-020-1601-5, 2021.

Yao, P., Yu, Z. G., Bianchi, T. S., Guo, Z. G., Zhao, M. X., Knappy, C. S., Keely, B. J., Zhao, B., Zhang, T. T., Pan, H. H., Wang, J. P., and Li, D.: A multiproxy analysis of sedimentary organic carbon in the Changjiang Estuary and adjacent shelf, J. Geophys. Res.-Biogeosci., 120, 1407-1429, https://doi.org/10.1002/2014jg002831, 2015.

Zhang, G. C., Liang, S. K., Shi, X. Y., and Wang, X. L.: Neutral Sugars in Sediment of the East Sea: Composition, Distribution, and Indication to Matter Degradation, Oceanologia et Limnologia Sinica, 45, 747-756 (in Chinese), 2014.

Zhou, F. X., Gao, X. L., Yuan, H. M., Song, J. M., and Chen, F. J.: The distribution and seasonal variations of sedimentary organic matter in the East China Sea shelf, Mar. Pollut. Bull., 129, 163-171, https://doi.org/10.1016/j.marpolbul.2018.02.009, 2018a.

Zhou, P., Song, X. X., Yuan, Y. Q., Cao, X. H., Wang, W. T., Chi, L. B., and Yu, Z. M.: Water Mass Analysis of the East China Sea and Interannual Variation of Kuroshio Subsurface Water Intrusion Through an Optimum Multiparameter Method, J. Geophys. Res.-Oceans, 123, 3723-3738, https://doi.org/10.1029/2018jc013882, 2018b.

Zhou, X. F.: Characterization and sources of sedimentary organic matter in Xingyun Lake, Jiangchuan, Yunnan, China, Environ. Earth Sci., 75, https://doi.org/10.1007/s12665-016-5853-5, 2016.

Zhu, C., Wagner, T., Pan, J. M., and Pancost, R. D.: Multiple sources and extensive degradation of terrestrial sedimentary organic matter across an energetic, wide continental shelf, Geochim. Cosmochim. Acta, 12, Q08011, https://doi.org/10.1029/2011gc003506, 2011.

Zhu, C., Wagner, T., Talbot, H. M., Weijers, J. W. H., Pan, J. M., and Pancost, R. D.: Mechanistic controls on diverse fates of terrestrial organic components in the East China Sea, Geochim. Cosmochim. Acta, 117, 129-143, https://doi.org/10.1016/j.gca.2013.04.015, 2013.

Zhu, C., Xue, B., Pan, J. M., Zhang, H. S., Wagner, T., and Pancost, R. D.: The dispersal of sedimentary terrestrial organic matter in the East China Sea (ECS) as revealed by biomarkers and hydro-chemical characteristics, Org. Geochem., 39, 952-957, https://doi.org/10.1016/j.orggeochem.2008.04.024, 2008.

---

## Author Comment (AC2)

**Response to reviewers of the manuscript entitled "Sources, Reactivity and Burial of Organic Matter in East China Sea Sediments, as Indicated by a Multi-geochemical Proxy Approach" authored by X. Yu, S. Liang, G. Zhang, S. Li, H. Huang and H. Ma.**

Below, the reviewer comments are included in blue, and responses in black font. All page/line numbers referenced below refer to the preprint.

**Reviewer #2**

**Comment 1:** In this study a set of surface samples, taken along a transect from the Changjiang outer estuary into the Okinawa Trough, were investigated with the main focus on the contribution of terrestrial (plant + non-plant) and marine organic matter. In addition to $\delta^{13}C$ and lignin which are used for end-member mixing, e.g. quantification of terrestrial vs. marine organic matter contribution, $\delta^{15}N$, TOC, TN, grain sizes, neutral sugar and amino acid content and spectral distributions are measured. This study is thus interesting due to its multi-proxy approach. A similar approach was used by Cowie, G., et al. (2014) (Comparative organic geochemistry of Indian margin (Arabian Sea) sediments: estuary to continental slope. Biogeosciences, 11(23), 6683–6696. doi:10.5194/bg-11-6683-2014.), and could thus be an important paper to compare the results with. It is interesting that despite a different major question the results are somewhat similar and stress the importance of grain size for organic matter accumulation. There may of course be more studies of a similar kind in the literature.

**Response:** Thank you for your comment and for providing the relevant reference. We have carefully read the suggested paper by Cowie et al. (2014) and agree that it provides a meaningful comparison in terms of methodological approach and findings. We will cite this reference in our manuscript (Part 4.1, Page 14, Line 322).

**Comment 2:** In the data presentation many of the results are shown but I really miss a Figure of the % terr as in Figure S2. This is a very informative Figure as it relates different variables such as clay content and % terr as well as lignin-phenol. Plant % would also be a good variable to be included in an additional Figure. The authors should include a Figure of the results of the end-member model with some of the relevant other variables.

**Response:** We will add a new Figure 10. Figure 10(b) further illustrates the outcomes of the end-member model alongside Clay (%), THAA (%TOC), and DI, thus addressing the reviewer's comments. In addition, Figure S2 was renamed as Figure 10(a). This

figure is as follows:

[Figure]

**Figure 10.** (a) Total organic carbon (TOC) content, percent (%) clay, % terrestrial organic matter ($f_{terr}$) and OC-normalized total lignin-phenol ($\Lambda 8$) concentration; (b) Carbon-normalized amino acid yields (THAA(%TOC)), % clay, % plant organic matter ($f_{plant}$), % non-plant organic matter ($f_{terr-plant}$), and % marine organic matter ($f_{marine}$) content, and degradation rate (DI) in sediments along the East China Sea study transect.

**Comment 3:** The introduction to neutral sugars and amino acid is quite short and should be enlarged with some reference to the use of these indicators (see comment to Lines 364-365).

**Response:** Following the reviewer's valuable suggestion, we will expand the introduction to include additional information on neutral sugars and amino acids, with particular emphasis on the use of Asp/Gly versus Ser+Thr as indicators. The relevant references will be added in Part 1, Page 3, Line 90, as follows:

"In addition, the aspartic acid (Asp)/glycine (Gly) ratio and serine (Ser) + threonine (Thr) (mol%) have been employed to differentiate between diatomaceous and calcareous sources of OM (Gupta and Kawahata, 2003; Ittekkot et al., 1984). Diatomaceous materials exhibit relatively lower Asp/Gly ratios (≈0.62%) than calcareous materials (≈1.88%), whereas calcareous materials are distinguished by relatively lower Ser + Thr (mol%) values (≈9%) than diatomaceous materials (≈16.7%) (Wei et al., 2022; Müller et al., 1986). Carbohydrates generally account for 3%-10% of TOC in marine sediments. NS are a class of monosaccharides within saccharides (i.e., carbohydrates), serving as important carbon and energy sources for microorganisms, with their yields typically declining during OM decomposition. The phytoplankton primary production is the main process controlling the distribution of carbohydrates on the shelf, while the high value of the relative percentage of fucose and rhamnose to total neutral sugars (mol% (Fuc+Rha)) indicates intensive diagenetic alteration (He et al., 2010)."

**Comment 4:** Lines 73-75: In contrast to the previous introduction to $\delta^{13}C$, the introduction to $\delta^{15}N$ is rather short. In addition, the range of terrestrial $\delta^{15}N$ values is much wider and a $\delta^{15}N$ of 0 ‰ is certainly not representative. A very comprehensive overview is available by "Kendall, C., Elliott, E. M. & Wankel, S. D. (2007). Tracing anthropogenic inputs of nitrogen to ecosystems, in *Stable Isotopes in Ecology and Environmental Science*. edited by R. H. Michener and K. Lajtha, pp. 375–449, Blackwell Publishing." But there are many more studies which suggest that less polluted and large rivers would have a $\delta^{15}N$ signal of about 4 ‰, whereas polluted rivers can have a much higher $\delta^{15}N$ and pristine mountain rivers a lower $\delta^{15}N$ value than 4 ‰ (Voss, M., Deutsch, B., Elmgren, R., Humburg, C., Kuuppoo, P., Pastuszak, M., Rolff, C. & Schulte, U. (2006). River biogeochemistry and source identification of nitrate by means of isotopic tracers in the Baltic Sea catchments. Biogeosciences Discussions, 3, 475–511.).

**Response:** We have carefully considered your suggestions and incorporated relevant information accordingly. Specifically, we will cite the two referenced articles and expanded the discussion on $\delta^{15}N$ in the Introduction section (Part 1, Page 3, Line 74) as follows:

"The range of $\delta^{15}N$ also suffers from the issue of overlapping isotopic signals, as values of algae and other aquatic plants range from -15‰ to 20‰, a range that encompasses those of terrestrial plants (-5‰ to 2‰), animal excreta (5‰), and soils (0.65‰ and 2.73‰ for cultivated and uncultivated soils, respectively) (Kendall et al., 2007). Isotopic $\delta^{15}N$ values are influenced by multiple processes. Biologically mediated reactions, including nitrogen fixation, assimilation, mineralization, nitrification, and denitrification, commonly result in an increase in the $\delta^{15}N$ values of substrates and a decrease in those of corresponding products (Kendall et al., 2007). Meanwhile, human activities also markedly influence the magnitude of $\delta^{15}N$ values. As noted by Voss et al. (2006), the $\delta^{15}N$ signal of large rivers with relatively little pollution is ca. 4‰, while that of polluted rivers may be much higher than this value, and the $\delta^{15}N$ values of pristine mountain rivers may be lower than 4‰."

**Comment 5:** Line 90: AA are not really molecular biomarkers as they are not very specific indicators of certain organisms but are ubiquitous in living organisms. They have, therefore, often been called biogeochemical indicators as they delineate degradation pathways (as the authors also describe).

**Response:** In accordance with the comment, we have revised the manuscript to reflect that amino acids (AAs) are better described as biogeochemical indicators rather than molecular biomarkers, given their ubiquity across living organisms and lack of

specificity. Specifically, the following changes have been made:

Part 1, Page 3, Line 90: "molecular biomarkers" has been replaced with "biogeochemical indicators".

**Comment 6:** Lines 118-120: a short description of the impact of currents on the study area is missing rather than just reporting their presence.

**Response:** A brief description of the impact of currents on the study area will be added in the Introduction section, Part 2.1, Page 4, Line 120, as follows:

"The YSCC flows southward throughout the year but is unable to penetrate the ECS during summer. In winter, however, it extends deeply into the northern ECS (Hwang et al., 2014). Together with the Yellow Sea Warm Current (YSWC), the YSCC forms a circulation system that induces upwelling, thus transporting nutrient-rich waters from deeper layers to the surface and enhancing primary productivity (Wang et al., 2019). The CDW contributes significantly to the nutrient pool of the ECS. During summer, most river-borne sediments are temporarily deposited in the subaqueous delta and estuarine zones. In winter, CDW flows southwestward along the coast toward the Taiwan Strait, and, in conjunction with the southward-flowing ZFCC, facilitates the transport of resuspended sediments (Hwang et al., 2014; Hu et al., 2014). The TWC flows northward year-round along the 50–100 m isobaths. In summer, the TWC originates from warm water in the Taiwan Strait and Kuroshio subsurface water, whereas in winter, it is primarily sustained by the Kuroshio subsurface intrusion, which reaches as far north as the subaqueous valley off the Changjiang Estuary (CJE) (Lian et al., 2016). Both the OKBC and the nearshore Kuroshio Branch Current (NKBC) deliver nutrient-enriched waters, particularly in phosphorus (P), thereby promoting primary productivity (Yang et al., 2012; Yang et al., 2013). The OKBC, flowing at depths of 60–120 m, moves northwestward along the 100 m isobath and partially rejoins the Kuroshio Main Stream near 28°N. The NKBC, originating at depths of 120–250 m, turns upward at approximately 27.5°N, 122°E, and then flows northeastward along the 60 m isobath until it reaches 30.5°N before veering eastward (Yang et al., 2012; Yang et al., 2013) (Fig. 1)."

**Comment 7:** Line 178: C/N instead of N/C

**Response:** We have corrected the expression from "N/C" to "C/N".

**Comment 8:** Line 300: Glu is missing in the Figure caption of Figure 5.

**Response:** We have revised the order of amino acids in the Figure 5 caption, placing Glu at the end to make it more prominent.

[Figure]

**Figure 5.** Concentration of each individual hydrolysable amino acid (HAA) (a) and the molar percentages of individual HAAs out of the total (THAA) (b) in surface sediments at sampling stations in an East China Sea transect (see Table S2). Gly: glycine, Asp: aspartic acid, Ala: alanine, Ser: serine, Val: valine, His: histidine, Thr: threonine, Leu: leucine, Arg: arginine, Ile: isoleucine, Phe: phenylalanine, Tyr: tyrosine and Glu: glutamic acid. Tyr concentration was too low to be visible in the plot.

**Comment 9:** Lines 323-324: imprecise: clay minerals are possibly not the sinks for AA but clay minerals may have organic coatings or fine organic matter may be transported with the clay fraction?

**Response:** We agree that the original statement was imprecise and will revise it accordingly. The sentence in Part 4.1, Page 14, Line 323: "clay may be the main sink for THAA and NS." will be replaced with a more accurate expression: "clay minerals may have organic coatings or fine organic matter (THAA and NS) may be transported with the clay fraction."

**Comment 10:** Line 331-333: The increase of $\delta^{15}N$ from the prodelta in offshore direction could be due to a reduced contribution of terrestrial material (see remark above).

**Response:** We thank the reviewer for this valuable comment. In response, we will revise the discussion accordingly and added relevant references. Specifically, we will supplement the discussion in Part 4.1, Page 14, Line 332 as follows:

"The increase of $\delta^{15}N$ from the prodelta in offshore direction could be due to a reduced contribution of terrestrial material (Kendall et al., 2007; Voss et al., 2006)."

**Comment 11:** Lines 364-365: the use of Asp/Gly vs. Ser+Thr as a source indicator

should be explained with reference in the methods section or introduction.

**Response:** The rationale for using Asp/Gly versus Ser+Thr as a source indicator will be explained and supported with references in the Introduction (Part 1, Page 3, Line 90). We will supplement the Introduction as follows:

"In addition, the aspartic acid (Asp)/glycine (Gly) ratio and serine (Ser) + threonine (Thr) (mol%) of THAA have been employed to differentiate between diatomaceous and calcareous sources of OM (Gupta and Kawahata, 2003; Ittekkot et al., 1984). Diatomaceous materials exhibit relatively lower Asp/Gly ratios (≈0.62%) than calcareous materials (≈1.88%), whereas calcareous materials are distinguished by relatively lower Ser + Thr (mol%) values (≈9%) than diatomaceous materials (≈16.7%) (Wei et al., 2022; Müller et al., 1986)."

**Comment 12:** Lines 421ff: When the TOC, TN and THAA contents are related to grain size and sorting rather than degradation processes it is feasible that THAA% and DI are not related as the material degrades as it is transported offshore or to greater depths. Further, the much lower SR and the decrease in offshore direction match the degradation as reflected in the DI. These aspects are discussed later but some of the discussion can be deleted by combining these aspects.

**Response:** Thank you for your insightful comment. In response, we will revise the original sentence "This is likely because each index is most sensitive to a particular diagenetic stage.   $DI_{AA}$ was most effective in reflecting diagenetic alterations during intermediate stages of decomposition over timescales of years to decades, whereas THAA(%TOC) and THAA(%TN) were more sensitive indicators of early stages of OM degradation stages (Davis et al., 2009; Chen et al., 2018)." to "This is likely because when the TOC, TN, and THAA contents are related to grain size and sorting rather than degradation processes, it is feasible that THAA(%TOC), THAA(%TN), and DI are not correlated as the material degrades during offshore transport or deposition at greater depths (Fig. 10(b)). Further, the much lower SR and the decrease in offshore direction match the degradation as reflected in the DI. ".

We have corrected the sentence "and the percentage of Gly increased (Fig. 5(b)) while that of Glc decreased" to "and the percentage of Gly (Fig. 5(b)) and Fuc+Rha (Fig. 4(b)) increased" in Line 418 of page 17 in part 4.2. The sentence "The mol% Glc concentration decreased seaward, except at DHa-5 (Fig. 4(b))." also was corrected to "The mol% (Fuc+Rha) increased seaward (Fig. 4(b))." in Line 277 of page 12 in part 3.3.2.

**Comment 13:** Line 465: see earlier comments on "molecular biomarkers" and change

the term here too.

**Response:** The term "molecular biomarkers" has been revised accordingly throughout the manuscript.

**Reference**

Chen, Y., Yang, G. P., Ji, C. X., Zhang, H. H., and Zhang, P. Y.: Sources and degradation of sedimentary organic matter in the mud belt of the East China Sea: Implications from the enantiomers of amino acids, Org. Geochem., 116, 51-61, https://doi.org/10.1016/j.orggeochem.2017.11.011, 2018.

Cowie, G., Mowbray, S., Kurian, S., Sarkar, A., White, C., Anderson, A., Vergnaud, B., Johnstone, G., Brear, S., Woulds, C., Naqvi, S. W. A., and Kitazato, H.: Comparative organic geochemistry of Indian margin (Arabian Sea) sediments: estuary to continental slope, Biogeosci., 11, 6683-6696, https://doi.org/10.5194/bg-11-6683-2014, 2014.

Davis, J., Kaiser, K., and Benner, R.: Amino acids and amino sugar yields and compositions as indicators of dissolved organic matter diagenesis, Org. Geochem., 40, 343-352, https://doi.org/10.1016/j.orggeochem.2008.12.003, 2009.

Gupta, L. P. and Kawahata, H.: Amino acids and hexosamines in the Hess Rise core during the past 220,000 years, Quat. Res., 60, 394-403, https://doi.org/10.1016/j.yqres.2003.07.012, 2003.

He, B., Dai, M., Huang, W., Liu, Q., Chen, H., and Xu, L.: Sources and accumulation of organic carbon in the Pearl River Estuary surface sediment as indicated by elemental, stable carbon isotopic, and carbohydrate compositions, Biogeosci., 7, 3343-3362, https://doi.org/10.5194/bg-7-3343-2010, 2010.

Hu, B., Li, J., Zhao, J., Wei, H., Yin, X., Li, G., Liu, Y., Sun, Z., Zou, L., Bai, F., Dou, Y., Wang, L., and Sun, R.: Late Holocene elemental and isotopic carbon and nitrogen records from the East China Sea inner shelf: Implications for monsoon and upwelling, Mar. Chem., 162, 60-70, https://doi.org/10.1016/j.marchem.2014.03.008, 2014.

Hwang, J. H., Van, S. P., Choi, B.-J., Chang, Y. S., and Kim, Y. H.: The physical processes in the Yellow Sea, Ocean Coastal Manage., 102, 449-457, https://doi.org/10.1016/j.ocecoaman.2014.03.026, 2014.

Ittekkot, V., Deuser, W. G., and Degens, E. T.: Seasonality in the fluxes of sugars, amino acids, and amino sugars to the deep ocean: Sargasso sea, Deep-Sea Res. Part I-Oceanogr. Res. Pap., 31, 1057-1069, https://doi.org/10.1016/0198-0149(84)90012-8, 1984.

Kendall, C., Elliott, E. M., and Wankel, S. D.: Tracing Anthropogenic Inputs of Nitrogen to Ecosystems, in: Stable Isotopes in Ecology and Environmental Science, edited by: Michener, R. H., and Lajtha, K., Blackwell Publishing, Oxford, 375-449, https://doi.org/abs/10.1002/9780470691854.ch12, 2007.

Lian, E., Yang, S., Wu, H., Yang, C., Li, C., and Liu, J. T.: Kuroshio subsurface water

feeds the wintertime Taiwan Warm Current on the inner East China Sea shelf, J. Geophys. Res.-Oceans, 121, 4790-4803, https://doi.org/10.1002/2016JC011869, 2016.

Müller, P. J., Suess, E., and AndréUngerer, C.: Amino acids and amino sugars of surface particulate and sediment trap material from waters of the Scotia Sea, Deep-Sea Res. Part I-Oceanogr. Res. Pap., 33, 819-838, https://doi.org/10.1016/0198-0149(86)90090-7 1986.

Voss, M., Deutsch, B., Elmgren, R., Humborg, C., Kuuppo, P., Pastuszak, M., Rolff, C., and Schulte, U.: Source identification of nitrate by means of isotopic tracers in the Baltic Sea catchments, Biogeosci., 3, 663-676, https://doi.org/10.5194/bg-3-663-2006, 2006.

Wang, Z., Xiao, X., Yuan, Z., Wang, F., Xing, L., Gong, X., Kubota, Y., Uchida, M., and Zhao, M.: Air-sea interactive forcing on phytoplankton productivity and community structure changes in the East China Sea during the Holocene, Glob. Planet. Change, 179, 80-91, https://doi.org/10.1016/j.gloplacha.2019.05.008, 2019.

Wei, J. E., Chen, Y., Zhang, N., Yang, J. Q., Chen, R., Zhang, H. H., and Yang, G. P.: Variability and composition of amino acids and amino sugars in sediment cores of the Changjiang Estuary, Org. Geochem., 163, 104330, https://doi.org/10.1016/j.orggeochem.2021.104330, 2022.

Yang, D., Yin, B., Sun, J., and Zhang, Y.: Numerical study on the origins and the forcing mechanism of the phosphate in upwelling areas off the coast of Zhejiang province, China in summer, J. Mar. Syst., 123, 1-18, https://doi.org/10.1016/j.jmarsys.2013.04.002, 2013.

Yang, D. Z., Yin, B. S., Liu, Z. L., Bai, T., Qi, J. F., and Chen, H. Y.: Numerical study on the pattern and origins of Kuroshio branches in the bottom water of southern East China Sea in summer, J. Geophys. Res.-Oceans, 117, c02014, https://doi.org/10.1029/2011jc007528, 2012.